# Neur2RO:
# Neural Two-Stage Robust Optimization

**Justin Dumouchelle***
University of Toronto

**Esther Julien**
TU Delft

**Jannis Kurtz**
University of Amsterdam

**Elias B. Khalil**
University of Toronto

## Abstract

Robust optimization provides a mathematical framework for modeling and solving decision-making problems under worst-case uncertainty. This work addresses two-stage robust optimization (2RO) problems (also called *adjustable robust optimization*), wherein first-stage and second-stage decisions are made before and after uncertainty is realized, respectively. This results in a nested min-max-min optimization problem which is extremely challenging computationally, especially when the decisions are discrete. We propose Neur2RO, an efficient machine learning-driven instantiation of column-and-constraint generation (CCG), a classical iterative algorithm for 2RO. Specifically, we learn to estimate the value function of the second-stage problem via a novel neural network architecture that is easy to optimize over by design. Embedding our neural network into CCG yields high-quality solutions quickly as evidenced by experiments on two 2RO benchmarks, knapsack and capital budgeting. For knapsack, Neur2RO finds solutions that are within roughly 2% of the best-known values in a few seconds compared to the three hours of the state-of-the-art exact branch-and-price algorithm; for larger and more complex instances, Neur2RO finds even better solutions. For capital budgeting, Neur2RO outperforms three variants of the $k$-adaptability algorithm, particularly on the largest instances, with a 10 to 100-fold reduction in solution time. Our code and data are available at https://github.com/khalil-research/Neur2RO.

## 1 Introduction

A wide range of real-world optimization problems in logistics, finance, and healthcare, among others, can be modeled by discrete optimization models (Petropoulos et al., 2023). While such mixed-integer (linear) problems (MILP) can still be challenging to solve, the problem size that can be tackled with modern solvers has increased significantly thanks to algorithmic developments (Wolsey, 2020; Achterberg & Wunderling, 2013). In recent years, the incorporation of Machine Learning (ML) models into established algorithmic frameworks has received increasing attention (Zhang et al., 2023; Bengio et al., 2021).

While most of ML for discrete optimization has focused on deterministic problems, in many cases, decision-makers face uncertainty in the problem parameters, e.g., due to forecasting or measurement errors in quantities of interest such as customer demand in inventory management. Besides the stochastic optimization approach, for which learning-based heuristics have been proposed recently (Dumouchelle et al., 2022), another popular approach to incorporate uncertainty into optimization models is *robust optimization*, where the goal is to find solutions which are optimal considering the worst realization of the uncertain parameters in a pre-defined uncertainty set (Ben-Tal et al., 2009). This more conservative approach has been extended to two-stage robust problems (2RO) where some of the decisions can be made on the fly after the uncertain parameters are realized (Ben-Tal et al., 2004); see Yanıkoğlu et al. (2019) for a survey.

**Example (Capital Budgeting).** As a classical example of a two-stage robust problem, consider the capital budgeting problem as defined in Subramanyam et al. (2020) where a company decides to invest in a subset of $n$ projects. Each project $i$ has an uncertain cost $c_i(\boldsymbol{\xi})$ and an uncertain profit

---

*Corresponding author: justin.dumouchelle@mail.utoronto.ca

$r_i(\boldsymbol{\xi})$ that both depend on the nominal cost and profit, respectively, and some risk factor $\boldsymbol{\xi}$ that dictates the difference from the nominal values to the actual ones. This risk factor, which we call an uncertain scenario, is contained in a given uncertainty set $\Xi$. The company can invest in a project either before or after observing the risk factor $\boldsymbol{\xi}$, up to a budget $B$. In the latter case, the company generates only a fraction $\eta$ of the profit, which reflects a penalty of postponement. The objective of the capital budgeting problem is to maximize the total revenue subject to the budget constraint. This problem can be formulated as:

$$\max_{\mathbf{x} \in \mathcal{X}} \min_{\boldsymbol{\xi} \in \Xi} \max_{\mathbf{y} \in \mathcal{Y}} \quad \mathbf{r}(\boldsymbol{\xi})^\mathsf{T}(\mathbf{x} + \eta\mathbf{y}) \tag{1a}$$

$$\text{s.t.} \qquad \mathbf{x} + \mathbf{y} \leq \mathbf{1} \tag{1b}$$

$$\mathbf{c}(\boldsymbol{\xi})^\mathsf{T}(\mathbf{x} + \mathbf{y}) \leq B. \tag{1c}$$

Here, $\mathcal{X} = \mathcal{Y} = \{0,1\}^n$ and $x_i$ and $y_i$ are the binary variables that indicate whether the company invests in the $i$-th project in the first- or second-stage, respectively. Constraint 1b ensures that the company can invest in each project only once and constraint 1c ensures that the total cost does not exceed the budget.

2RO with integer decisions is much harder to solve than deterministic MILPs, especially when the uncertain parameters appear in the constraints and the second-stage decisions are discrete. Even evaluating the objective value of a solution in this case is algorithmically challenging (Zhao & Zeng, 2012). In Subramanyam et al. (2020), none of the generated capital budgeting instances could be solved even approximately in a two-hour time limit for $n = 25$, terminating with an optimality gap of around $6\%$. In contrast to deterministic optimization problems, there is only limited literature on using ML methods to improve robust optimization (Julien et al., 2022; Goerigk & Kurtz, 2022).

**Contributions.** We propose Neural Two-stage Robust Optimization (`Neur2RO`), an ML framework that can quickly compute high-quality solutions for 2RO. Our contributions are as follows:

- **ML in a novel optimization setting:** 2RO (also known as *adjustable RO*) has been receiving increased interest from the operations research community (Yanıkoğlu et al., 2019) and our work is one of the first to leverage ML in this setting.

- **ML at the service of a classical optimization algorithm:** to deal with the highly constrained nature of real-world optimization problems and rather than attempting to predict solutions directly, we "neuralize" a well-established 2RO algorithm, a strategy that combines the best of both worlds: correctness of an established algorithm with the predictive capabilities of an accurate neural network.

- **A compact, generalizable neural architecture** that is MILP-representable and estimates the thorny component of a 2RO problem, namely the value of the second-stage problem. The network is invariant to problem size and parameters, allowing, for example, the use of the same architecture for capital budgeting instances with a different number of projects and budget parameters.

- **Competitive experimental results** on capital budgeting and a two-stage robust knapsack problem, both benchmarks in the 2RO literature. `Neur2RO` finds solutions that are of similar quality to or better than the state of the art. Large instances benefit the most from our method, with $100\times$ reduction and 10 to $100\times$ reductions in running time for knapsack and capital budgeting, respectively.

## 2 BACKGROUND

### 2.1 TWO-STAGE ROBUST OPTIMIZATION

2RO problems involve two types of decisions. The first set of decisions, $\mathbf{x}$, are referred to as *here-and-now* decisions and are made before the uncertainty is realized. The second set of decisions, $\mathbf{y}$, are referred to as the *wait-and-see* decisions and can be made on the fly after the uncertainty is realized. The uncertain parameters $\boldsymbol{\xi}$ are assumed to be contained in a convex and bounded uncertainty set $\Xi \subset \mathbb{R}^q$. The 2RO problem aims at finding a first-stage solution $\mathbf{x}$ which minimizes the worst-case objective value over all scenarios $\boldsymbol{\xi} \in \Xi$, where for each scenario the best possible

second-stage decision $\mathbf{y}(\boldsymbol{\xi})$ is implemented. Mathematically, a 2RO problem is given by

$$\min_{\mathbf{x}\in\mathcal{X}} \max_{\boldsymbol{\xi}\in\Xi} \min_{\mathbf{y}\in\mathcal{Y}} \quad \mathbf{c}(\boldsymbol{\xi})^{\mathsf{T}}\mathbf{x} + \mathbf{d}(\boldsymbol{\xi})^{\mathsf{T}}\mathbf{y} \tag{2a}$$

$$\text{s.t.} \quad T(\boldsymbol{\xi})\mathbf{x} + W(\boldsymbol{\xi})\mathbf{y} \leq \mathbf{h}(\boldsymbol{\xi}), \tag{2b}$$

where $\mathcal{X} \subseteq \mathbb{R}^n$ and $\mathcal{Y} \subseteq \mathbb{R}^m$ are feasible sets for the first and second stage decisions, respectively. In this work, we consider the challenging case of integer sets $\mathcal{X}$ and $\mathcal{Y}$. All parameters of the problem, namely $\mathbf{c}(\boldsymbol{\xi}) \in \mathbb{R}^n, \mathbf{d}(\boldsymbol{\xi}) \in \mathbb{R}^m, W(\boldsymbol{\xi}) \in \mathbb{R}^{r\times m}, T(\boldsymbol{\xi}) \in \mathbb{R}^{r\times n}$, and $\mathbf{h}(\boldsymbol{\xi}) \in \mathbb{R}^r$ depend on the scenario $\boldsymbol{\xi}$. We make the following assumption which is satisfied for the capital budgeting problem (and implicitly knapsack, which does not involve constraint uncertainty).

**Assumption.** For every $\mathbf{x} \in \mathcal{X}$, we have a method which calculates a scenario $\boldsymbol{\xi} \in \Xi$ for which the second-stage constraints $T(\boldsymbol{\xi})\mathbf{x} + W(\boldsymbol{\xi})\mathbf{y} \leq \mathbf{h}(\boldsymbol{\xi})$ over $\mathbf{y} \in \mathcal{Y}$ are infeasible or verifies that no such scenario exists.

Both single- and multi-stage robust mixed integer problems are NP-hard even for deterministic problems which can be solved in polynomial time Buchheim & Kurtz (2018). Compared to single-stage problems, which are often computationally tractable as they can be solved using reformulations Ben-Tal et al. (2009) or constraint generation Mutapcic & Boyd (2009), two-stage problems are much harder to solve. When dealing with integer first-stage and continuous recourse, CCG is one of the key approaches Zeng & Zhao (2013); Tsang et al. (2023). However, many problems, such as the ones we study here, deal with (mixed-)integer second-stage decisions. While an extension of CCG has been proposed that is able to handle mixed-integer recourse Zhao & Zeng (2012), this method is not well-established and often intractable and the results do not apply for pure integer second-stage problems.

In the case that the uncertainty only appears in the objective function, the 2RO can be solved by oracle-based branch-and-bound methods (Kämmerling & Kurtz, 2020), branch-and-price (Arslan & Detienne, 2022), or iterative cut generation using Fenchel cuts (Detienne et al., 2024). For special problem structures and binary uncertainty sets, a Lagrangian relaxation can be used to transform 2RO problems with constraint uncertainty into 2RO with objective uncertainty which can then be solved by the aforementioned methods (Subramanyam, 2022; Lefebvre et al., 2023).

## 2.2 Column-and-Constraint Generation

The main idea of CCG is to iterate between a *main problem* (MP) and an *adversarial problem* (AP). The MP is a relaxation of the original problem that only considers a finite subset of the uncertainty set $\Xi' \subset \Xi$. The latter problem can be modeled as a MILP by introducing copies of the second-stage decision variables $\mathbf{y}$ for each of the scenarios in $\Xi'$. After calculating an optimal solution of the MP, the AP finds new scenarios in the uncertainty set that cut off the current solution in the MP. When no such scenario can be found, the optimality of the current MP solution is guaranteed. For mathematical formulations of the two problems and a more detailed description of the CCG procedure, see Figure 1 and Appendix B.1.

CCG often fails to calculate an optimal solution in reasonable time since both the MP and the AP are very hard to solve. In each iteration, the size of the MP increases leading to it being difficult to solve to optimality even with commercial MILP solvers such as Gurobi (Gurobi Optimization, LLC, 2023). Furthermore, solving the AP is extremely challenging for integer second-stage variables. In Zhao & Zeng (2012), the authors present a column-and-constraint algorithm which solves the AP if the second stage is a mixed-integer problem; this leads to a CCG for the AP inside the main CCG, a most intractable combination. Additionally, the method of Zhao & Zeng (2012) is not applicable to purely integer second-stage decisions such as the problems we consider here.

## 3 Methodology

At a high level, our approach aims to train a neural network that predicts the optimal second-stage objective value function and then integrates this model within a CCG framework to obtain first-stage decisions. We rely on a training dataset of historical instances that can be used or generated, as is typically assumed in ML-for-optimization work.

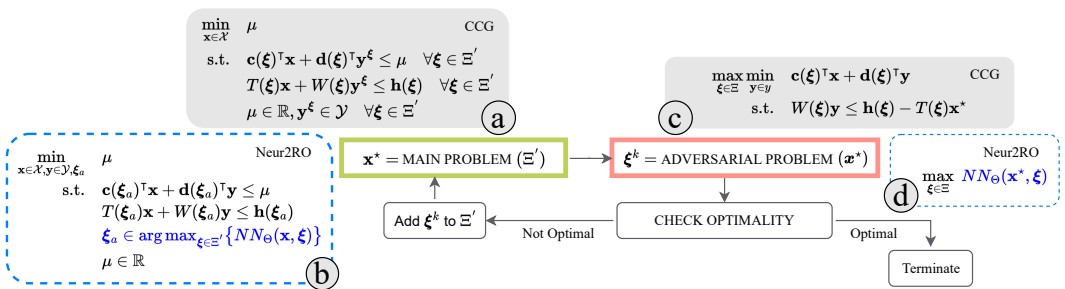

Figure 1: Column-and-constraint generation: in each iteration, a *main problem* (box (a)) is solved to find a good first-stage solution $\mathbf{x}^\star$ for the set of scenarios that have been identified thus far (initially, none). Then, an *adversarial problem* (box (c)) is solved to obtain a new scenario for which the solution $\mathbf{x}^\star$ is not feasible anymore in MP. If no such scenario exists, then $\mathbf{x}^\star$ is *optimal* and CCG terminates. Otherwise, the adversarial scenario is added to the set of worst-case scenarios and we iterate to MP. For each of the MP and AP, we show two versions: classical (CCG, boxes (a) and (c)) and learning-augmented (Neur2RO, dashed boxes (b) and (d)).

## 3.1 LEARNING MODEL

As mentioned before, CCG is computationally very expensive. Both the MP and AP contribute to its intractability (see Figure 1 boxes (a) and (c) for descriptions). In the MP, for each added scenario, a new second-stage decision $\mathbf{y}$ is introduced. When a large number of scenarios are required to obtain a robust solution, the number of variables grows rapidly. Moreover, the AP is especially hard when the second-stage decisions are integer, which is the case we consider. In our learning-augmented approach, we replace the intractable elements of the CCG with MILP representations of a trained NN which is computationally much easier to handle (Figure 1).

We train a neural network that can accurately predict the optimal value of the second-stage problem for a given input of a first-stage decision, an uncertainty realization, and the problem's specification, $\mathcal{P}$. The problem specification refers to the coefficients and size of the optimization problem, e.g., the nominal values of the profit and costs in the capital budgeting problem. More formally we train a neural network $NN_\Theta(\cdot)$, to approximate the optimal value of the integer problem

$$NN_\Theta(\mathbf{x}, \boldsymbol{\xi}, \mathcal{P}) \approx \min_{y \in \mathcal{Y}} \{\mathbf{c}_\mathcal{P}(\boldsymbol{\xi})^\mathsf{T}\mathbf{x} + \mathbf{d}_\mathcal{P}(\boldsymbol{\xi})^\mathsf{T}\mathbf{y} : W_\mathcal{P}(\boldsymbol{\xi})\mathbf{y} \le \mathbf{h}_\mathcal{P}(\boldsymbol{\xi}) - T_\mathcal{P}(\boldsymbol{\xi})\mathbf{x}\}, \qquad (3)$$

where $\Theta$ are the weights of the neural network. For ease of notation, we hereafter omit $\mathcal{P}$ in the formulation. For more details on the architecture of $NN_\Theta(\cdot)$, see Section 3.3. Alternatively, as $\mathbf{c}_\mathcal{P}(\boldsymbol{\xi})^\mathsf{T}\mathbf{x}$ is a scalar product of the input vectors, we instead could only predict the second-stage objective, i.e., $\mathbf{d}_\mathcal{P}(\boldsymbol{\xi})^\mathsf{T}\mathbf{y}$, subject to the same constraints. However, as demonstrated in Appendix H.3, predicting the sum of first- and second-stage objectives achieves higher-quality solutions.

## 3.2 ML-BASED COLUMN-AND-CONSTRAINT GENERATION

Having defined the learning task, we now describe the ML-based approximate CCG algorithm. For an overview of this method, see Figure 1.

**Main Problem.** Given a finite subset of scenarios $\Xi' \subset \Xi$, we reformulate the MP using an $\arg\max$ operator which selects a scenario that achieves the worst objective function value when replacing the second-stage objective value by the neural network formulation.

$$\min_{\mathbf{x} \in \mathcal{X}, \mathbf{y} \in \mathcal{Y}, \boldsymbol{\xi}_a \in \Xi} \quad \mathbf{c}(\boldsymbol{\xi}_a)^\mathsf{T}\mathbf{x} + \mathbf{d}(\boldsymbol{\xi}_a)^\mathsf{T}\mathbf{y} \qquad (4a)$$

$$\text{s.t.} \quad W(\boldsymbol{\xi}_a)\mathbf{y} + T(\boldsymbol{\xi}_a)\mathbf{x} \le \mathbf{h}(\boldsymbol{\xi}_a), \qquad (4b)$$

$$\boldsymbol{\xi}_a \in \arg\max_{\boldsymbol{\xi} \in \Xi'}\{NN_\Theta(\mathbf{x}, \boldsymbol{\xi})\}. \qquad (4c)$$

Modeling the $\arg\max$ can be done by adding additional linear constraints and binary variables, which we explicitly show in Appendix C. The MP results in a MILP formulation.

This formulation is indeed not the only option; for example, one could instead consider another formulation, called max, which provides a more intuitive formulation and does not require modeling second-stage variables; details are provided in Appendix H.1. However, equation 4 has one key property that motivates its efficacy. From the machine learning perspective, rather than requiring a neural network to be an accurate estimator of the objective for each scenario, we only require that the neural network be able to identify the maximal scenario. Prediction inaccuracy is then compensated for in equation 4a by exactly modeling the second-stage cost. As a result, when solving the MP, the true optimal first-stage decision for the selected scenario will be the minimizer, rather than a potentially suboptimal first-stage decision based on any inaccuracy of the learning model. Appendix H.1 presents an ablation comparing both the solution quality of the $\arg\max$ and max formulations on the knapsack problem, and establishes that the $\arg\max$ formulation indeed computes higher quality solutions across every instance.

**Adversarial Problem.** In the AP of `Neur2RO`, we replace the inner optimization problem over $\mathbf{y}$ by its prediction $NN_\Theta(\mathbf{x}^\star, \boldsymbol{\xi})$, where $\mathbf{x}^\star$ is given (see Figure 1 box (d)). When we deal with constraint uncertainty, we first check if there exists a scenario $\boldsymbol{\xi} \in \Xi$, such that no feasible $\mathbf{y} \in \mathcal{Y}$ exists for the constraints

$$W(\boldsymbol{\xi})\mathbf{y} + T(\boldsymbol{\xi})\mathbf{x}^\star \leq \mathbf{h}(\boldsymbol{\xi}),$$

which we can do by the assumption from Section 2.1. If such a scenario exists, we add it to $\Xi'$ and continue with solving the MP again. Note that in this case $\mathbf{x}^\star$ is not feasible for MP in the next iteration. If no such scenario could be found, we calculate an optimal solution $\boldsymbol{\xi}^\star$ of the AP in box (d) of Figure 1 which can be done by using the MILP representation for NN. Note that if $\Xi$ is a polyhedron or an ellipsoid, then this problem results in a mixed-integer linear or quadratic problem, respectively, which can be solved by state-of-the-art solvers such as Gurobi. We compare the optimal value of the latter problem with the objective values of all scenarios that were considered in the MP before. If the following holds

$$\max_{\boldsymbol{\xi} \in \Xi} NN_\Theta(\mathbf{x}^\star, \boldsymbol{\xi}) \geq \max_{\boldsymbol{\xi} \in \Xi'} NN_\Theta(\mathbf{x}^\star, \boldsymbol{\xi}) + \varepsilon \tag{5}$$

for a pre-defined accuracy parameter $\varepsilon > 0$, then we add $\boldsymbol{\xi}^\star$ to $\Xi'$ and continue with the MP. Otherwise, we stop the algorithm. Finally, note that we can calculate both types of scenarios in each iteration and add them both to $\Xi'$ before we iterate to the MP. As the adversarial problem requires finding the worst-case uncertainty over a neural network's input, heuristic approaches may significantly improve solving time with minimal degradation in solution quality. Appendix H.2 presents an ablation demonstrating significantly lower solution time at a minimal cost of solution quality for sampling- and relaxation-based heuristics.

**Convergence.** Since our algorithm does not apply the standard CCG steps, the convergence guarantee from the classical algorithm does not hold. However, we prove in Appendix F that it holds if only finitely many first-stage solutions exist, which is the case if all first-stage variables are integer and $\mathcal{X}$ is bounded; this indeed holds for the knapsack and capital budgeting problems.

**Theorem 1.** *If $\mathcal{X}$ is finite, the ML-based CCG terminates after a finite number of iterations.*

## 3.3 ARCHITECTURE

For the ML-based CCG, one requirement is the optimization, in each iteration, over several trained neural networks in the MP (one for each scenario in equation 4c) and a single trained neural network in the AP. Generally, increasing the size of the networks will lead to more challenging and potentially intractable optimization problems. For that reason, developing an architecture that can be efficiently optimized over is a crucial aspect of an efficient ML-based CCG algorithm.

To achieve efficient optimization, we embed the first-stage decisions, $\mathbf{x}$, and a scenario, $\boldsymbol{\xi}$, into low-dimensional embeddings using networks $\Phi_\mathbf{x}$ and $\Phi_{\boldsymbol{\xi}}$, respectively. These embeddings are concatenated and passed through a final small neural network (the *Value Network*) $\Phi$ that predicts the objective of the optimal second-stage response; see Figure 2 for a pictorial representation.

**Main Problem Optimization.** When representing the trained models in the MILP, we only have to represent the embedding network $\Phi_\mathbf{x}$ and the small value network $\Phi$, which can be done by classical MILP representations of ReLU NNs (Fischetti & Jo, 2018). Since the scenario parameters are not variables here, the scenario embeddings $\Phi_{\boldsymbol{\xi}}(\boldsymbol{\xi}^k)$ can be precomputed via a forward pass

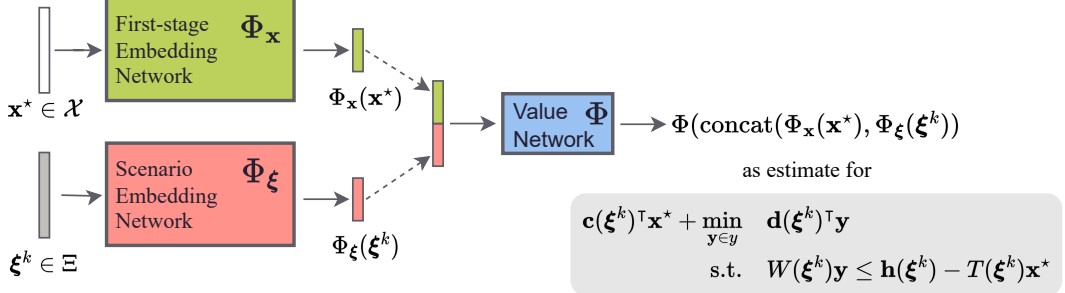

Figure 2: Neural network architecture for ML-based CCG. The current first-stage solution, $\mathbf{x}^\star$, is embedded once using the network $\Phi_{\mathbf{x}}(\cdot)$. A scenario $\boldsymbol{\xi}^k$ is embedded using the network $\Phi_{\boldsymbol{\xi}}(\cdot)$. To estimate the value of the second-stage optimization problem corresponding to a particular pair $(\mathbf{x}^\star, \boldsymbol{\xi}^k)$, the two embedding vectors are concatenated into one (dashed arrows) and then passed into the final *Value Network*.

for each scenario, i.e., no MILP representation is needed for $\Phi_{\boldsymbol{\xi}}$. If $\Phi$ is a small neural network, then representing a large number of copies of the network (one per scenario) remains amenable to efficient optimization.

**Adversarial Problem Optimization.** For the AP, we only require representing $\Phi_{\boldsymbol{\xi}}$ and $\Phi$ as the embedding of $\mathbf{x}$ can be precomputed with a forward pass.

**Generalizing Across Instances.** For simplicity of notation and presentation, the previous sections have omitted the generalization across instances, which is a key aspect of the generality of our methodology. To generalize across instances, invariance to the number, ordering, constraint coefficients, and objective coefficients of decision variables is required. To handle this, `Neur2RO` leverages set-based neural networks (Zaheer et al., 2017) for $\Phi_{\mathbf{x}}$ and $\Phi_{\boldsymbol{\xi}}$. Specifically, embeddings are computed for each single first-stage and scenario variable ($x_i$ and $\xi_i$) using their values, constraints, and objective coefficients, via a network with shared parameters. These embeddings are then aggregated and passed through an additional feed-forward neural network to derive the first-stage and scenario embeddings. For a detailed diagram of this architecture, see Appendix D.

## 4 EXPERIMENTAL SETUP

**Computational Setup.** All experiments were run on a computing cluster with an Intel Xeon CPU E5-2683 and Nvidia Tesla P100 GPU with 64GB of RAM (for training). Pytorch 1.12.1 (Paszke et al., 2019) was used for all learning models. Gurobi 10.0.2 (Gurobi Optimization, LLC, 2023) was used as the MILP solver and gurobi-machinelearning 1.3.0 was used to embed the neural networks into MILPs. For evaluation, all solving was limited to 3 hours. For `Neur2RO`, we terminate a solve of the MP or AP early if no improvement in solution is observed in 180 seconds. Our code and data are at `https://github.com/khalil-research/Neur2RO`.

**2RO Problems.** We benchmark `Neur2RO` on two 2RO problems from the literature, namely a two-stage knapsack problem and the capital budgeting problem. In both cases, our instances are as large or larger than considered in the literature. The two-stage knapsack problem is in the first stage a classical knapsack problem. The second stage has decisions for responding to an uncertain profit degradation. The capital budgeting problem is described in the introduction. For a detailed description of these problems, see Appendix A. Below we briefly detail each problem.

- **Knapsack.** For the knapsack problem, we use the same instances as in Arslan & Detienne (2022), which have been inspired by Ben-Tal et al. (2009). They have categorized their instances into four groups: uncorrelated (UN), weakly correlated (WC), almost strongly correlated (ASC), and strongly correlated (SC), which affects the correlation of the nominal profits of items with their cost and, in turn, the difficulty of the problem. More correlated instances are much harder to solve. We consider instances of sizes $n \in \{20, 30, 40, 50, 60, 70, 80\}$.

- **Capital budgeting.** These problem instances are generated similar to Subramanyam et al. (2020).While uncertain parameters appear in the constraints (see equation 1c), we can easily verify the assumption given in Section 2.1 as follows: for every $\mathbf{x}$ we check if $\max_{\boldsymbol{\xi} \in \Xi} \mathbf{c}(\boldsymbol{\xi}) \leq B$, where the maximum can be easily calculated since it is a linear problem over $\Xi$. If the latter inequality is true, the second-stage problem is feasible since we can choose $\mathbf{y} = \mathbf{0}$. On the other hand, if the inequality is violated, then no feasible second-stage solution exists since $\mathbf{y} \geq \mathbf{0}$, and hence the maximizing scenario will be added to MP. We consider instances of sizes $n \in \{10, 20, 30, 40, 50\}$.

**Baselines.** For knapsack, we compare to the branch-and-price (BP) algorithm from Arslan & Detienne (2022), the state-of-the-art for 2RO problems with objective uncertainty. We use the instances and the objective values and solution times reported in their paper (link). For capital budgeting, we use the $k$-adaptability approach of Subramanyam et al. (2020) (more details in Appendix B.2) with $k = 2, 5, 10$ as a baseline; CCG is not tractable for this problem due to its integer recourse.

**Evaluation.** After Neur2RO finds the first-stage decision $\mathbf{x}^\star$, we obtain the corresponding objective value by solving 2 for a fixed $\mathbf{x}$. For knapsack, this can be efficiently solved by constraint generation of $\mathbf{y}$. For capital budgeting on the other hand, due to constraint uncertainty, we cannot use the constraint generation approach. Instead we sample scenarios from $\Xi$, and solve 2 with fixed $\mathbf{x}$ and $\boldsymbol{\xi}$. See Appendix E for a more detailed explanation of these methods.

As previously mentioned, we use $k$-adaptability as the baseline for the capital budgeting problem. This method solves 2 only approximately with the approximation quality getting better with larger $k$ at an increase in solution times. We take the first-stage solution found by $k$-adaptability and compare it with the one of Neur2RO using the scenario sampling approach just described.

For the evaluation of the objective values, two metrics are considered. We use relative error (RE), i.e., the gap to the best-known solution, to compare solution quality. Specifically, if $obj^\star$ is the value of the best solution found by Neur2RO or a baseline for a particular instance, then for algorithm $A$ with objective $obj_A$, the RE is given by $100 \cdot \frac{|obj^\star - obj_A|}{|obj^\star|}$. To compare efficiency, we compare the average solution time.

**Data Collection & Training.** For data collection, we sample sets of instances, first-stage decisions, and scenarios to obtain features. The features are provided in Appendix I. Labels are then computed by solving the corresponding innermost optimization problem, i.e., a tractable deterministic MILP as both $\mathbf{x}$ and $\boldsymbol{\xi}$ are fixed. Additionally, this process is highly parallelizable since each optimization problem is independent. For knapsack and capital budgeting, we randomly sample 500 instances, 10 first-stage decisions per instance, and 50 scenarios per first-stage decision, resulting in 250,000 data points. The dataset is split into 200,000 and 50,000 samples for training and validation, respectively.

We train one size-independent model for each problem for 500 epochs. The data collection times, training times, and total times (in seconds) are 2,162, 3,789, and 5,951 for knapsack and 3,212, 2,195, and 5,407 for capital budgeting. We note that both times are relatively insignificant given that we provide approximately twice the time (3 hours) to solve a single instance during evaluation. Furthermore, the model for Neur2RO generalizes across instance parameters and sizes. Appendix I provides full detail on model hyperparameters and training.

## 5 EXPERIMENTAL RESULTS

For knapsack, we test our method and the baseline on 18 instances per correlation type and instance size (504 instances). For capital budgeting, we test on 50 instances per instance size (250 instances). We note that training and validation data are generated using the procedures specified in the corresponding papers, and different instances are used for testing. For optimization of Neur2RO, this section presents results for solving the MIP and MILP formulations for the MP and AP, with the $\arg\max$ formulation outlined in Section 3 for the MP. Tables 1-2 report the median RE and solving times. In addition, for more detailed distributional information, boxplots and more detailed metrics, are provided in Appendix G.

**Knapsack.** Table 1 demonstrates a clear improvement in scalability, with the solving time of Neur2RO ranging between 4 and 77 seconds, while the solving time for BP scales directly with difficulty induced by the size and correlation type. For the more difficult instances, i.e., instances

| Correlation Type | # items | Median RE | | Times | | | Correlation Type | # items | Median RE | | Times | |
|---|---|---|---|---|---|---|---|---|---|---|---|---|
| | | Neur2RO | BP | Neur2RO | BP | | | | Neur2RO | BP | Neur2RO | BP |
| Uncorrelated | 20 | 1.417 | **0.000** | 4 | **0** | | Almost Strongly Correlated | 20 | 1.439 | **0.000** | **5** | 9 |
| | 30 | 1.188 | **0.000** | 6 | **1** | | | 30 | 0.782 | **0.000** | **6** | 2708 |
| | 40 | 1.614 | **0.000** | 9 | **3** | | | 40 | 0.497 | **0.000** | **10** | 4744 |
| | 50 | 1.814 | **0.000** | **9** | 12 | | | 50 | 0.019 | **0.000** | **7** | 8852 |
| | 60 | 1.146 | **0.000** | **14** | 18 | | | 60 | **0.000** | 0.016 | **14** | 10261 |
| | 70 | 1.408 | **0.000** | **16** | 46 | | | 70 | **0.017** | 0.031 | **13** | 10800 |
| | 80 | 0.968 | **0.000** | **11** | 388 | | | 80 | **0.000** | 0.265 | **12** | 10800 |
| Weakly Correlated | 20 | 1.582 | **0.000** | **5** | 29 | | Strongly Correlated | 20 | 1.604 | **0.000** | **5** | 9 |
| | 30 | 2.236 | **0.000** | **11** | 454 | | | 30 | 0.610 | **0.000** | **7** | 2473 |
| | 40 | 1.595 | **0.000** | **20** | 6179 | | | 40 | 0.443 | **0.000** | **11** | 5665 |
| | 50 | 1.757 | **0.000** | **19** | 8465 | | | 50 | 0.073 | **0.000** | **9** | 8240 |
| | 60 | 0.695 | **0.000** | **77** | 9242 | | | 60 | 0.042 | **0.010** | **11** | 10800 |
| | 70 | 0.165 | **0.000** | **15** | 10800 | | | 70 | **0.020** | 0.027 | **16** | 10800 |
| | 80 | **0.000** | 0.341 | **21** | 10800 | | | 80 | **0.000** | 0.179 | **13** | 10800 |

Table 1: Median RE and solving times for knapsack instances. For each row, the median RE and average solving time are computed over 18 instances. All times in seconds. The smallest (best) values in each row/metric are in bold.

| # items | Median RE | | | | Times | | | |
|---|---|---|---|---|---|---|---|---|
| | Neur2RO | $k = 2$ | $k = 5$ | $k = 10$ | Neur2RO | $k = 2$ | $k = 5$ | $k = 10$ |
| 10 | 1.105 | 1.140 | **0.000** | **0.000** | 59 | **20** | 9561 | 10800 |
| 20 | **0.000** | 0.196 | 0.112 | 0.064 | **324** | 8702 | 10800 | 10800 |
| 30 | 0.109 | **0.020** | 0.073 | 0.032 | **602** | 10801 | 10800 | 10800 |
| 40 | **0.009** | 0.074 | 0.011 | 0.019 | **739** | 10806 | 10801 | 10801 |
| 50 | **0.001** | 0.033 | 0.039 | 0.020 | **1032** | 10807 | 10804 | 10801 |

Table 2: Combined results for capital budgeting instances. For each row, the median RE and average solving time are computed over 50 instances. All times in seconds. The smallest (best) values in each row/metric are in bold.

with a large number of items and (almost) strong correlation, Neur2RO generally finds better quality solutions over 100 times faster than BP, which is a very strong result considering BP is the state-of-the-art for problems with objective uncertainty. Figures 6-7 of Appendix G further demonstrate that the distribution of RE achieved by Neur2RO, not just the median, is far more favorable than BP's on the most challenging instances. For easier instances, Neur2RO is less competitive in terms of solution quality as BP converges to optimal solutions within the time limit. However, even for these instances, Neur2RO achieves a median RE of 2.235% in the worst-case, often still 1-2 orders of magnitude faster than BP, with the exception of a few very easy instances.

**Capital budgeting.** Neur2RO achieves the lowest median RE for 20, 40, and 50-item instances, i.e., the two largest and most challenging instance sets. The distribution of RE for 40 and 50-item instances provided in Figure 8 of Appendix G is indeed consistent with the median result, as it illustrates that Neur2RO finds quality solutions on the majority of the instances. In terms of solving time, Neur2RO generally converges much faster than $k$-adaptability, resulting in a very favorable trade-off: we can find better or equally good solutions 10 to 100 times faster. Note that the relative errors are quite small in an absolute sense. For example, for 30-item instances, Neur2RO has a median RE of 0.109 compared to the best baseline's 0.020; solutions that are within 0.109% of the best achievable may be acceptable in practice. Note that we have also measured the median RE for k-adaptability assuming a shorter time limit, namely the same amount of time as Neur2RO on each instance. Taking the incumbent solution found by k-adaptability at that time point typically yields worse solutions than those reported in Table 2, see Appendix H.4 for details. Compared to the knapsack, the solving time is generally much larger as the instance size increases. We speculate that this may relate to the uncertainty in the objective of the first-stage decision or the budget constraints that are not present in the knapsack problem.

In summary, for both benchmark problems, Neur2RO achieves high-quality solutions. For relatively easy or small instances, state-of-the-art methods sometimes find slightly better solutions, often at a much higher computational cost. However, as the instances become more difficult, Neur2RO demonstrates a clear improvement in overall solution quality and computing time.

## 6 RELATED WORK

**Robust optimization.** Besides the exact solution methods mentioned in Section 2.1, several heuristic methods have been developed to derive near-optimal solutions for mixed-integer 2RO problems. Methods that solve 2RO heuristically are $k$-adaptability (Bertsimas & Caramanis, 2010; Hanasusanto et al., 2015; Subramanyam et al., 2020), decision rules (Bertsimas & Georghiou, 2018; 2015), and iteratively splitting the uncertainty set (Postek & Hertog, 2016). Machine Learning techniques have been developed to speed up solution algorithms for the $k$-adaptability problem in Julien et al. (2022). In Goerigk & Kurtz (2022) a decision tree classifier is trained to predict good start scenarios for the CCG. While being heuristic solvers, all of the above methods are still computationally highly demanding. In this paper, the $k$-adaptability branch-and-bound algorithm by Subramanyam et al. (2020) is used as a baseline since it is one of the only methods which is able to calculate high quality solutions for reasonable problem sizes. For an elaborate overview of the latter algorithm, see Appendix B.2.

Besides improving algorithmic performance, ML methods have been used to construct uncertainty sets based on historical data. In Goerigk & Kurtz (2023) one-class neural networks are used to construct highly complex and non-convex uncertainty sets. Results from statistical learning theory are used to derive guarantees for ML designed uncertainty sets in Tulabandhula & Rudin (2014). Other approaches use principal component analysis and kernel smoothing (Ning & You, 2018), support vector clustering (Shang et al., 2017; Shang & You, 2019; Shen et al., 2020), statistical hypothesis testing (Bertsimas et al., 2018), or Dirichlet process mixture models (Ning & You, 2017; Campbell & How, 2015). In Wang et al. (2023a) uncertainty sets providing a certain probabilistic guarantee are derived by solving a CVaR-constrained bilevel problem by an augmented Lagrangian method. While interesting and related, we here assume the uncertainty set is known.

**MILP representations of neural networks.** One key aspect of `Neur2RO` is representing neural networks as constraints and variables in MILPs, which was first explored in Cheng et al. (2017); Tjeng et al. (2017); Fischetti & Jo (2018); Serra et al. (2018). These representations have motivated active research to improve the MILP solving efficiency of optimizing over-trained models Grimstad & Andersson (2019); Anderson et al. (2020); Wang et al. (2023b), as well as several software contributions Bergman et al. (2022); Ceccon et al. (2022), in addition to Gurobi, a commercial MILP solver, providing an open-source library. The use of embedding trained predictive models to derive approximate MILPs has been explored for non-linear constraints or intractable constraints (Say et al., 2017; Grimstad & Andersson, 2019; Murzakhanov et al., 2020; Katz et al., 2020; Kody et al., 2022), stochastic programming (Dumouchelle et al., 2022; Kronqvist et al., 2023), and bilevel optimization (Dumouchelle et al., 2024). As `Neur2RO` is based on an approximation for intractable 2RO problems with embedded neural networks, the latter area of research is the most closely related. However, the min-max-min optimization in 2RO renders previous learning-based MILP approximations unsuitable.

## 7 CONCLUSION

With the uncertainty in real-world noisy data, the economy, the climate, and other avenues, there is an increasing need for efficient robust decision-making. We have shown how `Neur2RO` uses MILP-representable feedforward neural networks to estimate the thorny component of a family of two-stage robust optimization instances, namely the value of the second-stage problem. The neural network architecture delicately combines low-dimensional embeddings of a first-stage decision and a scenario to produce the second-stage value estimate. Using an off-the-shelf MILP solver, we then use the neural network in a classical iterative algorithm for 2RO. `Neur2RO` to find competitive solutions compared to state-of-the-art methods on two challenging benchmark problems, knapsack and capital budgeting, at a substantial reduction in solution time.

Our work paves the way for further integration of learning and optimization under uncertainty. The AP in the CCG algorithm is over the inputs of a neural network, which could benefit from the many "adversarial attack" heuristics in the ML literature. Generalizations of `Neur2RO` that make predictions of the infeasibility of a particular constraint could be explored. ReLU networks are not the only class of models that is MILP-representable; decision tree models could be used as alternatives and may be appropriate for other problem settings.

ACKNOWLEDGEMENTS

Dumouchelle and Khalil acknowledge support from the Scale AI Research Chair Program and an NSERC Discovery Grant. Julien acknowledges funding from the Netherlands Organisation for Scientific Research (NWO).

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

# A 2RO PROBLEMS

## A.1 ROBUST TWO-STAGE KNAPSACK

We consider the two-stage knapsack problem as defined in Arslan & Detienne (2022) with a set of $n$ items. Each item $i$ has a weight $c_i$ and an uncertain profit $p_i(\boldsymbol{\xi}) = \bar{p}_i - \xi_i \hat{p}_i$, where $\bar{p}_i$ is the expected profit, $\hat{p}_i$ its maximum deviation and $\xi_i$ the *uncertain* profit degradation factor, where the degradation happens after the first stage. In this problem we have a budgeted uncertainty set $\Xi = \{\boldsymbol{\xi} \in [0,1]^n : \sum_{i=1}^n \xi_i \leq \Gamma\}$. The first stage decision is to choose a subset of items to produce. Then in the second stage, there are three different responses to the profit degradation: (i) accept the degraded profit, (ii) repair the item by using an additional $t_i$ units from the budget to recover the original profit $\bar{p}_i$, or (iii) outsource the item for a cost of $f_i$ units, such that the item's profit results in $\bar{p}_i - f_i$. This gives the following problem formulation:

$$\min_{\mathbf{x} \in \{0,1\}^n} \max_{\boldsymbol{\xi} \in \Xi} \min_{\mathbf{y} \in \{0,1\}^n, \mathbf{r} \in \{0,1\}^n} \sum_{i=1}^n (f_i - \bar{p}_i)x_i + (\hat{p}_i \xi_i - f_i)y_i - \hat{p}_i \xi_i r_i$$

$$\text{s.t.} \quad \sum_{i=1}^n c_i y_i + t_i r_i \leq C$$

$$r_i \leq y_i \leq x_i \qquad \qquad \forall i \in \{1, \dots, n\},$$

where $x_i$ is the first-stage decision to produce item $i$. For the second-stage decisions, we have $y_i$ and $r_i$: (i) $y_i = 1$ if item $i$ is produced *without* repairing and $y_i = 0$ if the item is outsourced, and (ii) $r_i$ is the decision for repairing item $i$.

## A.2 CAPITAL BUDGETING

Consider the capital budgeting problem in Subramanyam et al. (2020), where a company aims to invest in a subset of $n$ projects. For each project, $i$, the uncertain cost, and profit are respectively defined as

$$c_i(\boldsymbol{\xi}) = \left(1 + \boldsymbol{\Phi}_i^\mathsf{T} \boldsymbol{\xi}/2\right)\bar{c}_i \quad \text{and} \quad r_i(\boldsymbol{\xi}) = \left(1 + \boldsymbol{\Psi}_i^\mathsf{T} \boldsymbol{\xi}/2\right)\bar{r}_i, \quad \forall i \in \{1, \dots, n\},$$

where $\bar{c}_i$ and $\bar{r}_i$ are the nominal cost and nominal profit of project $i$. $\boldsymbol{\Phi}_i^\mathsf{T}$ and $\boldsymbol{\Psi}_i^\mathsf{T}$ are the $i$-th row vectors of the sensitivity matrices $\boldsymbol{\Phi}, \boldsymbol{\Psi} \in \mathbb{R}^{n \times 4}$, with $\boldsymbol{\xi} \in \Xi = [-1, 1]^4$. We use the problem formulation described in 1.

# B 2RO ALGORITHMS

In this section, we describe the column-and-constraint generation algorithm in more detail and the $k$-adaptability problem, briefly describing one of its solution methods.

## B.1 COLUMN-AND-CONSTRAINT GENERATION

The CCG iterates between the *main problem* and the *adversarial problem* (AP). The MP is given as

$$\min_{\mathbf{x} \in \mathcal{X}} \max_{\boldsymbol{\xi} \in \Xi'} \min_{y \in \mathcal{Y}} \quad \mathbf{c}(\boldsymbol{\xi})^\mathsf{T} \mathbf{x} + \mathbf{d}(\boldsymbol{\xi})^\mathsf{T} \mathbf{y} \tag{6a}$$

$$\text{s.t.} \quad T(\boldsymbol{\xi})\mathbf{x} + W(\boldsymbol{\xi})\mathbf{y} \leq \mathbf{h}(\boldsymbol{\xi}), \tag{6b}$$

where $\Xi' \subset \Xi$ is a finite subset of scenarios. Clearly, the MP provides a lower bound on the optimal value of equation 2. To solve the MP, for each scenario in $\Xi'$ a copy of the second-stage variables is generated. Using a level-set transformation, the problem can be formulated as

$$\min_{\mathbf{x} \in \mathcal{X}} \quad \mu \tag{7a}$$

$$\text{s.t.} \quad \mathbf{c}(\boldsymbol{\xi})^\mathsf{T} \mathbf{x} + \mathbf{d}(\boldsymbol{\xi})^\mathsf{T} \mathbf{y}^{\boldsymbol{\xi}} \leq \mu \quad \forall \boldsymbol{\xi} \in \Xi' \tag{7b}$$

$$T(\boldsymbol{\xi})\mathbf{x} + W(\boldsymbol{\xi})\mathbf{y}^{\boldsymbol{\xi}} \leq \mathbf{h}(\boldsymbol{\xi}) \quad \forall \boldsymbol{\xi} \in \Xi' \tag{7c}$$

$$\mu \in \mathbb{R}, \mathbf{y}^{\boldsymbol{\xi}} \in \mathcal{Y} \quad \forall \boldsymbol{\xi} \in \Xi', \tag{7d}$$

which is a linear integer problem that state-of-the-art solvers, such as Gurobi, can solve. In each iteration of the CCG an optimal solution $(x^*, \mu^*)$ of equation 7 is calculated. Afterwards, the AP is solved, which is defined as

$$\max_{\boldsymbol{\xi} \in \Xi} \min_{\mathbf{y} \in \mathcal{Y}} \quad \mathbf{c}(\boldsymbol{\xi})^\mathsf{T} \mathbf{x}^\star + \mathbf{d}(\boldsymbol{\xi})^\mathsf{T} \mathbf{y} \tag{8a}$$

$$\text{s.t.} \quad W(\boldsymbol{\xi})\mathbf{y} \leq \mathbf{h}(\boldsymbol{\xi}) - T(\boldsymbol{\xi})\mathbf{x}^\star. \tag{8b}$$

Since the optimal value of the AP is the objective value of the current solution $\mathbf{x}^*$, it provides an upper bound on the optimal value of equation 2. We define the optimal value to be equal to infinity if there exists a scenario $\boldsymbol{\xi} \in \Xi$ for which no feasible second-stage solution $\mathbf{y}$ exists. If the optimal value of the AP is larger than $\mu^*$ then we add the optimal scenario $\xi^*$ to $\Xi'$ and start again from solving MP. Otherwise, we stop the algorithm since the upper bound is smaller or equal to the lower bound, and hence $\mathbf{x}^*$ is an optimal solution. The whole procedure is presented in Algorithm 1.

---

**Algorithm 1** Column-and-Constraint Generation

---

set $ub = \infty$, $lb = -\infty$
$\Xi' = \{\xi_0\}$ for any $\xi_0 \in \Xi$
**while** $ub - lb > 0$ **do**
    Calculate an optimal solution $x^*, \mu^*$ of the main problem equation 7 and set $lb = \mu^*$.
    Calculate an optimal solution $\xi^*$ (with optimal value $opt^*$) of the adversarial problem equation 8 where $x = x^*$.
    Set $\Xi' = \Xi' \cup \{\xi^*\}$ and $ub = \min\{ub, opt^*\}$.
**end while**
**return** $x^*$

---

CCG often fails to calculate an optimal solution in a reasonable time since both the MP and the AP are very hard to solve in the case of integer second-stage variables. In each iteration, the size of MP increases since we have to add new constraints and a copy of all integer second-stage decisions $\mathbf{y}$. This often leads to the situation that after even a small number of iterations, the MP cannot be solved to optimality anymore by classical integer optimization solvers as Gurobi.

Furthermore, solving the AP is extremely challenging for integer second-stage variables. Indeed, the problem can be formulated as a bilevel problem where the follower problem contains integer variables. In Zhao & Zeng (2012) the authors present a column-and-constraint algorithm that solves the AP if the second-stage is a mixed-integer problem. One drawback is that this method is not applicable if the second-stage does not contain continuous variables, as is the case for many problems, e.g., the capital budgeting problem. Furthermore, the method involves solving a very large mixed-integer bilinear problem, which is computationally enormously challenging. The whole procedure must be executed in each iteration of the main CCG algorithm.

## B.2 $k$-ADAPTABILITY

The $k$-adaptability approach was introduced in Bertsimas & Caramanis (2010) and later studied for objective uncertainty and constraint uncertainty in Hanasusanto et al. (2015); Subramanyam et al. (2020); Ghahtarani et al. (2023); Julien et al. (2022); Kurtz (2023). The main idea of the approach is to calculate a set of $k$ second-stage solutions already in the first-stage. Instead of choosing the best feasible second-stage solution for each scenario $\boldsymbol{\xi}$, we choose the best of the $k$ calculated second-stage solutions. Since we restrict the number of second-stage reactions, this approach leads to feasible solutions of equation 2, which are not necessarily optimal. While for larger $k$ the approximation guarantee gets provably better, the problem gets harder to solve at the same time. Furthermore, it was shown in Subramanyam et al. (2020) that it may happen that $k$ has to be chosen exponentially large to guarantee optimality for equation 2. The $k$-adaptability problem can be formulated as

$$\min_{\mathbf{x} \in \mathcal{X}, \mathbf{y}^1, \ldots, \mathbf{y}^k \in \mathcal{Y}} \max_{\boldsymbol{\xi} \in \Xi} \min_{y \in \{y^1, \ldots, y^k\}} \quad \mathbf{c}(\boldsymbol{\xi})^\mathsf{T} \mathbf{x} + \mathbf{d}(\boldsymbol{\xi})^\mathsf{T} \mathbf{y} \tag{9a}$$

$$\text{s.t.} \quad W(\boldsymbol{\xi})\mathbf{y} + T(\boldsymbol{\xi})\mathbf{x} \leq \mathbf{h}(\boldsymbol{\xi}). \tag{9b}$$

The $k$-adaptability problem is very challenging to solve, especially in the constraint uncertainty case. The best-known method for this case was introduced in Subramanyam et al. (2020). The authors perform a branch-and-bound algorithm over partitions of the uncertainty set. They consider $k$-partitions of finite scenarios sets, which are iteratively generated, and assign each of the second-stage solutions to one of the partitions. This approach was later improved by applying machine learning methods to improve the branching decisions Julien et al. (2022). As an alternative approach in Postek & Hertog (2016), an iterative uncertainty set splitting method is presented, which converges to the exact optimal value of the two-stage robust problem.

In case of objective uncertainty, the $k$-adaptablity problem is easier (but still hard) to solve (Arslan et al. (2022); Ghahtarani et al. (2023)) and can be approximated if $k$ is not too small; see Kurtz (2023).

## C  DETAILED FORMULATION

This section presents the detailed $\arg\max$ formulation for equation 4. We assume that at this iteration in the MP, we have scenarios $\boldsymbol{\xi}_1, \ldots, \boldsymbol{\xi}_k$ and that $M$ and $L$ are upper and lower bounds on the prediction of the network. The complete formulation is then given by

$$\min_{\mathbf{x}\in\mathcal{X}, \mathbf{y}\in\mathcal{Y}, \boldsymbol{\xi}_a\in\Xi, \mathbf{p}, u, \mathbf{z}\in\{0,1\}^k} \quad \mathbf{c}(\boldsymbol{\xi}_a)^\mathsf{T}\mathbf{x} + \mathbf{d}(\boldsymbol{\xi}_a)^\mathsf{T}\mathbf{y} \tag{10a}$$

$$\text{s.t.} \quad W(\boldsymbol{\xi}_a)\mathbf{y} + T(\boldsymbol{\xi}_a)\mathbf{x} \leq \mathbf{h}(\boldsymbol{\xi}_a), \tag{10b}$$

$$p_i = NN_\Theta(\mathbf{x}, \boldsymbol{\xi}_i) \qquad \forall i \in \{1, \ldots, k\} \tag{10c}$$

$$u \geq p_i \qquad \forall i \in \{1, \ldots, k\} \tag{10d}$$

$$u \leq p_i + (M - L)(1 - z_i) \qquad \forall i \in \{1, \ldots, k\} \tag{10e}$$

$$\sum_{i=1}^k z_i = 1 \qquad \forall i \in \{1, \ldots, k\} \tag{10f}$$

$$\boldsymbol{\xi}_a = \sum_{i=1}^k z_i \cdot \boldsymbol{\xi}_i \tag{10g}$$

To model the $\arg\max$, we introduce $k$ binary variables $\mathbf{z}$ and $k+1$ continuous variables $\mathbf{p}$ and $u$, which are used to model big-$M$ that ensure $\mathbf{z}$ is 1 at the index of the maximizer and 0 everywhere else. $\boldsymbol{\xi}_a$ is then given by a linear combination of the scenarios multiplied with $\mathbf{z}$.

## D  EXTENDED NN ARCHITECTURE

We show the extended neural network architecture used in the experiments in Figure 3.

## E  2RO WITH FIXED FIRST-STAGE DECISION

When we compare the calculated solutions of `Neur2RO` and the baseline in our experiments, we need to calculate the objective value of a solution $\mathbf{x}^\star \in \mathcal{X}$ exactly or approximately. The former involves solving the AP equation 8 for a given solution. Solving this problem is intractable when we have uncertain parameters in the constraints. We first expand on how the adversarial would be solved in a tractable way if the uncertain parameters only appear in the objective function. Subsequently, we describe an approach to approximately solve the AP, which is based on sampling scenarios from $\Xi$.

### E.1  OBJECTIVE UNCERTAINTY

For the special case of objective uncertainty, the AP can be solved much more efficiently. In this case, the adversarial problem is given as

$$\max_{\boldsymbol{\xi}\in\Xi} \min_{\mathbf{y}\in\mathcal{Y}} \quad \mathbf{c}(\boldsymbol{\xi})^\mathsf{T}\mathbf{x}^\star + \mathbf{d}(\boldsymbol{\xi})^\mathsf{T}\mathbf{y} \tag{11a}$$

$$\text{s.t.} \quad W\mathbf{y} \leq \mathbf{h} - T\mathbf{x}^\star, \tag{11b}$$

Figure 3: The extended neural network architecture for ML-based CCG. Compared to the NN architecture shown in the main text (Figure 2), this model uses the set-based method to be able to generalize across instance sizes. Let $\mathbf{x}^\star \in \mathbb{R}^n$ and $\boldsymbol{\xi} \in \mathbb{R}^q$. Then, $\hat{\Phi}_{\mathbf{x}}$ and $\hat{\Phi}_{\boldsymbol{\xi}}$ are the embedding networks for $x_i, i \in [n]$ and $\xi_j, j \in [q]$, respectively. The features are comprised of the single variable and single-variable specific problem specifications $\mathcal{P}_{\mathbf{x}}^i, i \in [n]$ and $\mathcal{P}_{\boldsymbol{\xi}}^j, j \in [q]$ for first-stage decisions and scenarios, respectively. The outputs of the $\hat{\Phi}$ networks are aggregated for $\mathbf{x}^*$ and $\boldsymbol{\xi}$ separately. These embeddings are the input of the original NN given in the main part.

which can be reformulated as

$$\max_{\boldsymbol{\xi} \in \Xi} \quad \alpha \tag{12a}$$

$$\text{s.t.} \quad \alpha \le \mathbf{c}(\boldsymbol{\xi})^\mathsf{T}\mathbf{x}^\star + \mathbf{d}(\boldsymbol{\xi})^\mathsf{T}\mathbf{y} \quad \forall \mathbf{y} \in \bar{\mathcal{Y}}, \tag{12b}$$

where $\bar{\mathcal{Y}} = \{\mathbf{y} \in \mathcal{Y}: \quad W\mathbf{y} \le \mathbf{h} - T\mathbf{x}^\star\}$. While the set $\bar{\mathcal{Y}}$ can contain an exponential number of solutions, the latter problem can be solved by iteratively generating the constraints for $y \in \bar{\mathcal{Y}}$.

### E.2 Constraint Uncertainty

We collect all scenarios $\xi \in \Xi$ which were generated during training and during the solution procedures of the baseline algorithm and our algorithm (including the scenarios calculated by the AP) in the set $\Xi^{samples}$. Then for the two returned solutions $\mathbf{x}^*$ and $\mathbf{x}^{baseline}$ we compare

$$\max_{\boldsymbol{\xi} \in \Xi^{samples}} \min_{\mathbf{y} \in \mathcal{Y}} \quad \mathbf{c}(\boldsymbol{\xi})^\mathsf{T}\mathbf{x}^\star + \mathbf{d}(\boldsymbol{\xi})^\mathsf{T}\mathbf{y} \tag{13a}$$

$$\text{s.t.} \quad W(\boldsymbol{\xi})\mathbf{y} \le \mathbf{h}(\boldsymbol{\xi}) - T(\boldsymbol{\xi})\mathbf{x}^\star, \tag{13b}$$

where we replace $\mathbf{x}$ by the corresponding solution $\mathbf{x}^*$ or $\mathbf{x}^{baseline}$. The latter problem can be solved by calculating the optimal value of the second-stage problem for each scenario independently and choosing the worst-case overall optimal values.

## F Convergence

In the following, we present the proof of Theorem 1.

*Proof.* The main idea is to show that the condition equation 5 cannot hold in infinitely many iterations. Since we stop the algorithm if 5 is not true anymore, then finite termination of the algorithm follows.

Assume the algorithm does not terminate in a finite number of iterations. Let $l_t$ and $r_t$ be the values of the left-hand side and right-hand side of inequality 5 in iteration $t$ of the algorithm, i.e.,

$$l_t := \max_{\boldsymbol{\xi} \in \Xi} NN_\Theta(\mathbf{x}^t, \boldsymbol{\xi})$$

and

$$r_t := \max_{\boldsymbol{\xi} \in \Xi^t} NN_\Theta(\mathbf{x}^t, \boldsymbol{\xi}).$$

where $\mathbf{x}^t$ is the optimal solution of MP in the $t$-th iteration and $\Xi^t$ the finite set of scenarios used in the MP in iteration $t$. Let $\mathbf{x} \in \mathcal{X}$ be a feasible first-stage solution and let $l_t(\mathbf{x})$ and $r_t(\mathbf{x})$ be the sub-sequences which contain the values of $l_t$ and $r_t$ only for the iterations where $x$ is an optimal solution of the MP. Then either this sequence is finite or, if it is infinite, the sequence $\{r_t(\mathbf{x})\}_t$ is monotonous and bounded where monotony follows since $\Xi^t \subset \Xi^{t+1}$ and since the same $x$ is used. The sequence is bounded since $\Xi$ is a bounded set and $NN_\Theta$ a piecewise-linear function (as is known for feedforward ReLU networks (Montufar et al., 2014)) and the maximum of a piecewise linear function over a bounded set is bounded. Hence, $\{r_t(x)\}_t$ converges to a finite value $r^\star(\mathbf{x})$. Furthermore, it holds $l_t(\mathbf{x}) \leq r_{t+1}(\mathbf{x})$ since the optimal scenario of the left-hand-side is added to $\Xi^t$ which is a subset of the set later used to evaluate $r_{t+1}(\mathbf{x})$. It follows that

$$r_t(\mathbf{x}) \leq l_t(\mathbf{x}) - \varepsilon \leq r_{t+1}(\mathbf{x}) - \varepsilon$$

for all $t$ which contradicts the convergence of $r_t(\mathbf{x})$. Hence the sequence $r_t(\mathbf{x})$ must be finite. Since only finitely many first-stage solutions $\mathbf{x}$ exist, and the latter result holds for all of them, the number of iterations of the algorithm must be finite. □

## G   DISTRIBUTIONAL RESULTS FOR RELATIVE PERFORMANCE

In this section, we provide distributional information for the RE for knapsack in Tables 3-4 and Figures 4-8.

| Correlation Type | # items | Mean RE Neur2RO | BP | Median RE Neur2RO | BP | RE 1st Quartile Neur2RO | BP | RE 3rd Quartile Neur2RO | BP |
|---|---|---|---|---|---|---|---|---|---|
| Uncorrelated | 20 | 2.005 | **0.000** | 1.417 | **0.000** | 0.541 | **0.000** | 2.379 | **0.000** |
| | 30 | 1.189 | **0.000** | 1.188 | **0.000** | 0.712 | **0.000** | 1.399 | **0.000** |
| | 40 | 2.895 | **0.000** | 1.614 | **0.000** | 1.221 | **0.000** | 4.042 | **0.000** |
| | 50 | 3.032 | **0.000** | 1.814 | **0.000** | 0.946 | **0.000** | 3.801 | **0.000** |
| | 60 | 2.099 | **0.000** | 1.146 | **0.000** | 0.577 | **0.000** | 2.872 | **0.000** |
| | 70 | 2.214 | **0.000** | 1.408 | **0.000** | 0.761 | **0.000** | 2.506 | **0.000** |
| | 80 | 1.591 | **0.000** | 0.968 | **0.000** | 0.758 | **0.000** | 2.063 | **0.000** |
| Weakly Correlated | 20 | 2.569 | **0.000** | 1.582 | **0.000** | 1.229 | **0.000** | 4.010 | **0.000** |
| | 30 | 2.664 | **0.000** | 2.236 | **0.000** | 0.616 | **0.000** | 4.293 | **0.000** |
| | 40 | 2.320 | **0.000** | 1.595 | **0.000** | 1.164 | **0.000** | 2.292 | **0.000** |
| | 50 | 2.183 | **0.145** | 1.757 | **0.000** | 0.793 | **0.000** | 2.674 | **0.000** |
| | 60 | 2.165 | **0.390** | 0.695 | **0.000** | **0.000** | 0.000 | 3.445 | **0.458** |
| | 70 | 0.884 | **0.338** | 0.165 | **0.000** | **0.000** | 0.000 | 0.623 | **0.175** |
| | 80 | **0.392** | 0.691 | **0.000** | 0.341 | **0.000** | 0.000 | **0.165** | 0.831 |
| Almost Strongly Correlated | 20 | 2.355 | **0.000** | 1.439 | **0.000** | 0.000 | 0.000 | 2.757 | **0.000** |
| | 30 | 1.166 | **0.113** | 0.782 | **0.000** | 0.075 | **0.000** | 1.911 | **0.000** |
| | 40 | 0.825 | **0.335** | 0.497 | **0.000** | 0.019 | **0.000** | 1.606 | **0.000** |
| | 50 | **0.314** | 0.884 | 0.019 | **0.000** | **0.000** | 0.000 | **0.229** | 1.251 |
| | 60 | **0.197** | 0.523 | **0.000** | 0.016 | **0.000** | 0.000 | **0.268** | 1.129 |
| | 70 | **0.551** | 0.615 | **0.017** | 0.031 | **0.000** | 0.000 | **1.058** | 1.227 |
| | 80 | **0.388** | 0.694 | **0.000** | 0.265 | **0.000** | 0.000 | **0.554** | 0.770 |
| Strongly Correlated | 20 | 2.387 | **0.000** | 1.604 | **0.000** | 0.905 | **0.000** | 3.018 | **0.000** |
| | 30 | 1.068 | **0.121** | 0.610 | **0.000** | 0.054 | **0.000** | 1.939 | **0.000** |
| | 40 | 0.658 | **0.191** | 0.443 | **0.000** | 0.002 | **0.000** | 0.888 | **0.000** |
| | 50 | **0.411** | 0.648 | 0.073 | **0.000** | **0.000** | 0.000 | **0.780** | 0.963 |
| | 60 | **0.322** | 0.367 | 0.042 | **0.010** | **0.000** | 0.000 | **0.173** | 0.693 |
| | 70 | **0.389** | 0.738 | **0.020** | 0.027 | **0.000** | 0.000 | **0.535** | 0.793 |
| | 80 | **0.318** | 0.668 | **0.000** | 0.179 | **0.000** | 0.000 | **0.245** | 0.906 |

Table 3: Table of distributional information for knapsack. For each row, all RE statistics are computed over 18 instances.

| # items | Mean RE Neur2RO | $k=2$ | $k=5$ | $k=10$ | Median RE Neur2RO | $k=2$ | $k=5$ | $k=10$ | RE 1st Quartile Neur2RO | $k=2$ | $k=5$ | $k=10$ | RE 3rd Quartile Neur2RO | $k=2$ | $k=5$ | $k=10$ |
|---|---|---|---|---|---|---|---|---|---|---|---|---|---|---|---|---|
| 10 | 2.558 | 2.849 | **1.029** | 1.165 | 1.105 | 1.140 | **0.000** | **0.000** | **0.000** | **0.000** | **0.000** | **0.000** | 3.534 | 4.349 | **0.547** | 1.557 |
| 20 | 0.423 | 0.304 | **0.232** | 0.266 | **0.000** | 0.196 | 0.112 | 0.064 | **0.000** | 0.094 | 0.013 | **0.000** | 0.410 | 0.453 | **0.320** | 0.362 |
| 30 | 0.408 | 0.149 | 0.131 | **0.084** | 0.109 | **0.020** | 0.073 | 0.032 | 0.002 | **0.000** | 0.003 | **0.000** | 0.337 | 0.182 | 0.212 | **0.110** |
| 40 | 0.234 | 0.114 | 0.098 | **0.073** | **0.009** | 0.074 | 0.011 | 0.019 | **0.000** | 0.001 | **0.000** | 0.002 | **0.121** | 0.180 | 0.137 | 0.137 |
| 50 | 0.090 | 0.107 | 0.090 | **0.056** | **0.001** | 0.033 | 0.039 | 0.020 | **0.000** | 0.000 | **0.000** | 0.002 | **0.050** | 0.193 | 0.139 | 0.084 |

Table 4: Table of distributional information for capital budgeting. For each row, all RE statistics are computed over 50 instances.

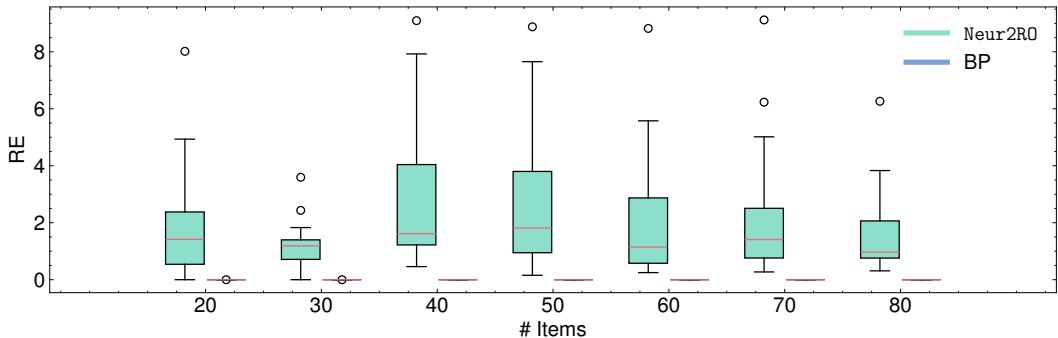

Figure 4: Boxplot of RE for baseline and `Neur2RO` on UN knapsack instances.

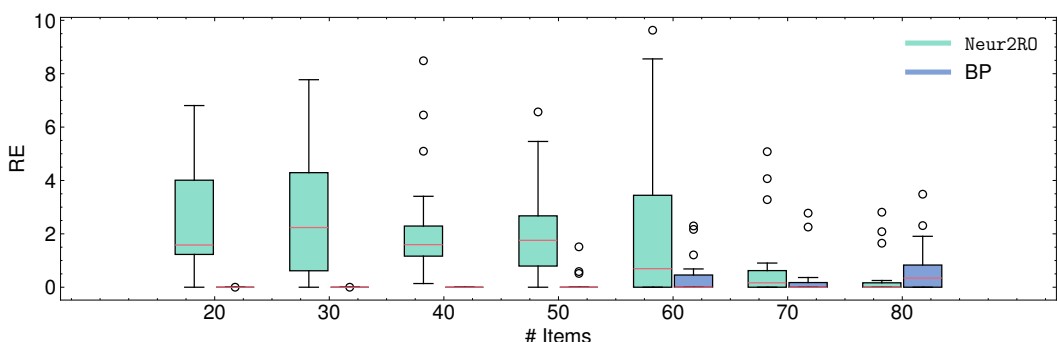

Figure 5: Boxplot of RE for baseline and `Neur2RO` on WC knapsack instances.

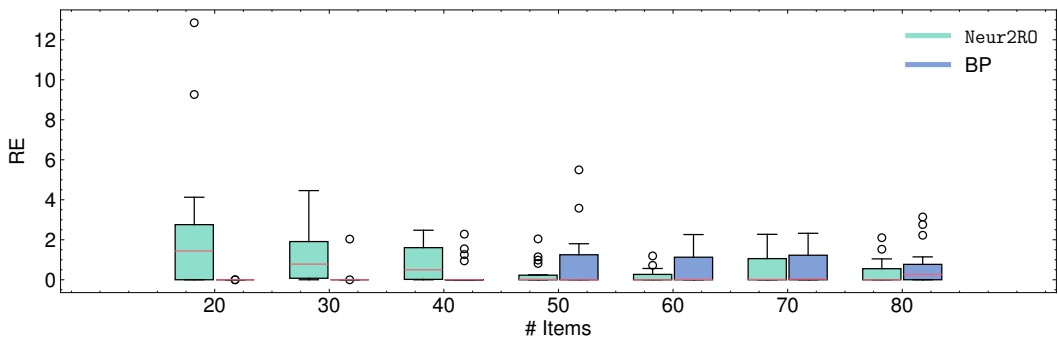

Figure 6: Boxplot of RE for baseline and `Neur2RO` on ASC knapsack instances.

# H    ABLATION

This section presents an ablation across two aspects of `Neur2RO`, namely, the formulation of the MP and the method to obtain worst-case scenarios. Both results are presented on the knapsack instances.

## H.1    MAIN PROBLEM FORMULATION

As an alternative to the formulation using $\arg\max$ over a set of scenarios. One more straightforward formulation is to consider instead the max over all of the scenarios, which is given by

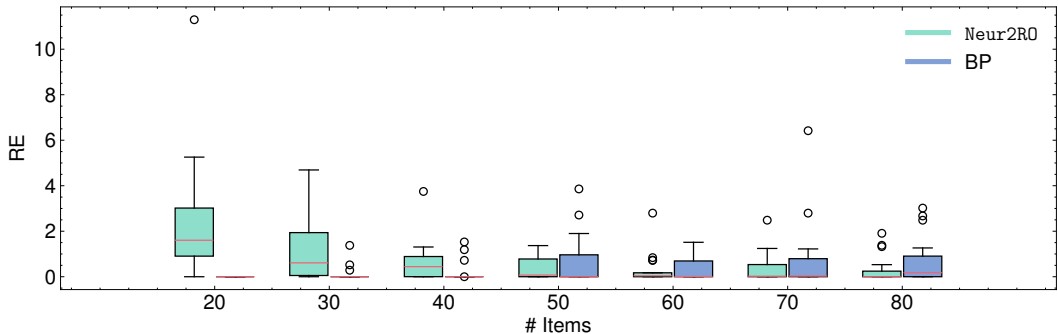

Figure 7: Boxplot of RE for baseline and `Neur2RO` on SC knapsack instances.

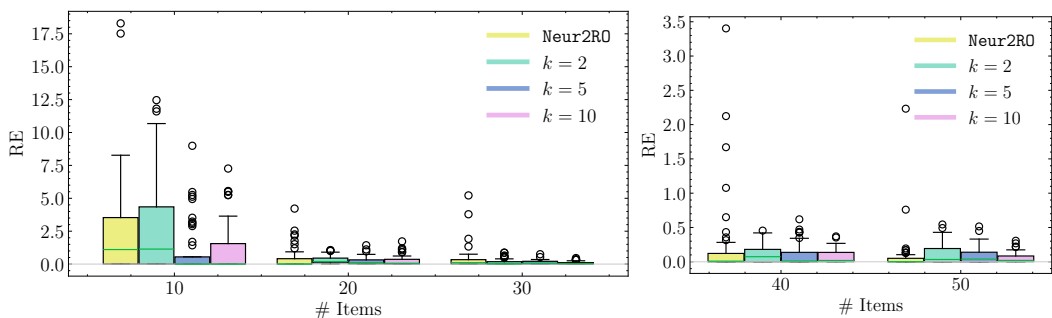

Figure 8: Box plot of RE for baselines and `Neur2RO` on capital budgeting instances.

$$\min_{\mathbf{x}\in\mathcal{X},\alpha} \quad \alpha \tag{14a}$$

$$\text{s.t.} \quad \alpha \geq NN_\Theta(\mathbf{x},\boldsymbol{\xi}_i) \qquad\qquad \forall k \in \{1,\ldots,K\}. \tag{14b}$$

Table 5 reports the MRE of the $\arg\max$ and $\max$ formulations and the solving time. Table 5 demonstrates an improvement in solution quality, with $\arg\max$ obtaining a lower MRE in every case and a lower computing time in most cases.

## H.2 WORST-CASE SCENARIO ACQUISITION

This section compares the adversarial approach for determining scenarios to a sampling and a linear programming (LP) relaxation-based approach.

### H.2.1 SAMPLING-BASED SCENARIO ACQUISITION

For sampling, as a baseline, we sample $100,000$ scenarios, and then to approximate the AP, we take the maximizer over a forward pass. Table 6 demonstrates a clear trade-off between solution quality and efficiency. Generally, sampling improves average solving time across all instances but leads to worse solution quality as the instance size increases.

### H.2.2 LP RELAXATION-BASED SCENARIO ACQUISITION

For 2RO, the uncertainty set is often polyhedral, which scenarios can be heuristically obtained via a LP relaxation. For the LP relaxation, we compare the performance of the standard MILP-based scenario acquisition (standard), i.e., solving the AP to optimality, to the relaxation (LP relaxation). For both problems, we report the RE to the baselines. Tables 7 and 8 present the knapsack and

| Correlation Type | # items | Median RE | | Times | |
|---|---|---|---|---|---|
| | | arg max | max | arg max | max |
| Uncorrelated | 20 | **0.000** | 1.167 | **5** | 11 |
| | 30 | **0.000** | 0.945 | **7** | 14 |
| | 40 | **0.000** | 1.931 | **9** | 24 |
| | 50 | **0.000** | 1.634 | **10** | 33 |
| | 60 | **0.000** | 0.452 | **17** | 29 |
| | 70 | **0.000** | 0.801 | **19** | 28 |
| | 80 | **0.000** | 2.227 | **13** | 35 |
| Weakly Correlated | 20 | **0.000** | 3.515 | **6** | 13 |
| | 30 | **0.000** | 2.405 | **11** | 22 |
| | 40 | **0.000** | 0.502 | **26** | 42 |
| | 50 | **0.000** | 0.254 | **24** | 39 |
| | 60 | **0.000** | 1.528 | 77 | **58** |
| | 70 | **0.000** | 1.769 | **18** | 35 |
| | 80 | **0.000** | 3.492 | **27** | 75 |
| Almost Strongly Correlated | 20 | **0.000** | 2.042 | **5** | 12 |
| | 30 | **0.000** | 1.433 | **6** | 14 |
| | 40 | **0.000** | 1.739 | **11** | 33 |
| | 50 | **0.000** | 3.161 | **8** | 20 |
| | 60 | **0.000** | 2.449 | **15** | 30 |
| | 70 | **0.000** | 2.497 | **18** | 35 |
| | 80 | **0.000** | 1.824 | **17** | 30 |
| Strongly Correlated | 20 | **0.000** | 1.154 | **5** | 11 |
| | 30 | **0.000** | 0.967 | **7** | 15 |
| | 40 | **0.000** | 1.928 | **16** | 28 |
| | 50 | **0.000** | 3.613 | **10** | 21 |
| | 60 | **0.000** | 2.005 | **20** | 26 |
| | 70 | **0.000** | 2.657 | **16** | 33 |
| | 80 | **0.000** | 2.051 | **16** | 28 |

Table 5: $\arg\max$ and $\max$ formulations on knapsack instances. For each row, the median RE and solving time are computed over 18 instances. All times in seconds.

capital budgeting results, respectively. In general, we can observe that the LP relaxation leads to significantly faster solving time, with an overall decreased solution quality. That being said, for capital budgeting in particular, `Neur2RO` with the LP relaxation still achieves a lower median RE than the baselines on larger instances, while being roughly five times faster than results without the relaxation.

## H.3 Prediction Target

This section compares the prediction target. For capital budgeting, the coefficients of the first-stage decisions in the objective contain uncertainty. As such, this presents a choice of either predicting the sum of the first- and second-stage objectives, i.e., $\mathbf{c}(\boldsymbol{\xi})^\mathsf{T}\mathbf{x} + \min_{\mathbf{y}\in\mathcal{Y}}\left\{\mathbf{d}(\boldsymbol{\xi})^\mathsf{T}\mathbf{y} : W(\boldsymbol{\xi})\mathbf{y} \leq \mathbf{h}(\boldsymbol{\xi}) - T(\boldsymbol{\xi})\mathbf{x}\right\}$, or only the second-stage objective, i.e., $\min_{\mathbf{y}\in\mathcal{Y}}\left\{\mathbf{d}(\boldsymbol{\xi})^\mathsf{T}\mathbf{y} : W(\boldsymbol{\xi})\mathbf{y} \leq \mathbf{h}(\boldsymbol{\xi}) - T(\boldsymbol{\xi})\mathbf{x}\right\}$. Specifically, we compare the downstream optimization performance with respect to the resulting formulations. The formulation for predicting the sum of the first- and second-stage objectives is presented in Section 3. For predicting the second-stage objective only, the MP is given by

$$\min_{\mathbf{x}\in\mathcal{X},\mathbf{y}\in\mathcal{Y},\boldsymbol{\xi}_a\in\Xi} \quad \mathbf{c}(\boldsymbol{\xi}_a)^\mathsf{T}\mathbf{x} + \mathbf{d}(\boldsymbol{\xi}_a)^\mathsf{T}\mathbf{y} \tag{15a}$$

$$\text{s.t.} \quad W(\boldsymbol{\xi}_a)\mathbf{y} + T(\boldsymbol{\xi}_a)\mathbf{x} \leq \mathbf{h}(\boldsymbol{\xi}_a), \tag{15b}$$

$$\boldsymbol{\xi}_a \in \arg\max_{\boldsymbol{\xi}\in\Xi'}\left\{\mathbf{c}(\boldsymbol{\xi})^\mathsf{T}\mathbf{x} + NN_\Theta(\mathbf{x},\boldsymbol{\xi})\right\}, \tag{15c}$$

and the AP is given by

$$\max_{\boldsymbol{\xi}\in\Xi} \mathbf{c}(\boldsymbol{\xi})^\mathsf{T}\mathbf{x}^\star + NN_\Theta(\mathbf{x}^\star,\boldsymbol{\xi}). \tag{16}$$

The main difference with this formulation is that the objective coefficients $\mathbf{c}(\boldsymbol{\xi})$ can be utilized directly rather than requiring the ML model to predict them. Table 9 compares the two approaches on the capital budgeting instances wherein the RE is computed with respect to the baselines. Empirically, we can see that predicting the sum of the first- and second-stage objectives yields significantly better solutions. On the methodological side, when only the second stage is predicted each node in the branch-and-bound tree being explored by a MIP solver will contain the exact first-stage and the predicted second-stage objectives. As such, we speculate that the LP relaxation at each node will

| Correlation Type | # items | Median RE | | Times | |
|---|---|---|---|---|---|
| | | adversarial | sampling | adversarial | sampling |
| Uncorrelated | 20 | **0.000** | **0.000** | 5 | **2** |
| | 30 | **0.000** | **0.000** | 7 | **4** |
| | 40 | 0.560 | **0.000** | 9 | **4** |
| | 50 | 0.723 | **0.000** | 10 | **5** |
| | 60 | 0.066 | **0.000** | 17 | **6** |
| | 70 | 0.150 | **0.000** | 19 | **8** |
| | 80 | 0.395 | **0.000** | 13 | **9** |
| Weakly Correlated | 20 | **0.000** | 0.074 | 6 | **3** |
| | 30 | **0.000** | 0.444 | 11 | **4** |
| | 40 | **0.000** | 0.093 | 26 | **5** |
| | 50 | 0.441 | **0.000** | 24 | **7** |
| | 60 | 0.119 | **0.065** | 77 | **9** |
| | 70 | **0.000** | 0.185 | 18 | **8** |
| | 80 | **0.000** | 0.536 | 27 | **9** |
| Almost Strongly Correlated | 20 | **0.000** | **0.000** | **5** | 5 |
| | 30 | **0.000** | **0.000** | **6** | 6 |
| | 40 | **0.000** | **0.000** | 11 | **10** |
| | 50 | **0.000** | **0.000** | 8 | **7** |
| | 60 | **0.000** | **0.000** | 15 | **14** |
| | 70 | **0.000** | **0.000** | 18 | **13** |
| | 80 | **0.000** | **0.000** | 17 | **12** |
| Strongly Correlated | 20 | **0.000** | **0.000** | 5 | **5** |
| | 30 | **0.000** | **0.000** | 7 | **7** |
| | 40 | **0.000** | **0.000** | 16 | **11** |
| | 50 | **0.000** | **0.000** | 10 | **8** |
| | 60 | **0.000** | **0.000** | 20 | **13** |
| | 70 | **0.000** | **0.000** | 16 | **14** |
| | 80 | **0.000** | **0.000** | 16 | **13** |

Table 6: Adversarial and sampling-based approaches for worst-case scenario acquisition on knapsack instances. For each row, the median RE and solving time are computed over 50 instances. All times in seconds.

| Correlation Type | # items | Median RE | | Times | |
|---|---|---|---|---|---|
| | | standard | LP relaxation | standard | LP relaxation |
| Uncorrelated | 20 | **1.417** | 1.673 | 4 | **1** |
| | 30 | 1.188 | **1.167** | 6 | **1** |
| | 40 | 1.614 | **1.387** | 9 | **2** |
| | 50 | 1.814 | **1.660** | 9 | **2** |
| | 60 | **1.146** | **1.146** | 14 | **1** |
| | 70 | 1.408 | **1.166** | 16 | **2** |
| | 80 | 0.986 | **0.970** | 11 | **2** |
| Weakly Correlated | 20 | 1.582 | **1.454** | 5 | **1** |
| | 30 | 2.236 | **2.034** | 11 | **1** |
| | 40 | **1.595** | 2.733 | 20 | **2** |
| | 50 | 1.757 | **1.126** | 19 | **2** |
| | 60 | **0.695** | 0.729 | 77 | **3** |
| | 70 | **0.165** | 0.243 | 15 | **3** |
| | 80 | **0.000** | 0.316 | 21 | **9** |

| Correlation Type | # items | Median RE | | Times | |
|---|---|---|---|---|---|
| | | standard | LP relaxation | standard | LP relaxation |
| Almost Strongly Correlated | 20 | 1.439 | **1.211** | 5 | **1** |
| | 30 | 0.782 | **0.665** | 6 | **1** |
| | 40 | **0.497** | 0.927 | 10 | **2** |
| | 50 | **0.019** | 1.884 | 7 | **2** |
| | 60 | **0.000** | 1.079 | 14 | **2** |
| | 70 | **0.017** | 0.025 | 13 | **4** |
| | 80 | **0.000** | 1.775 | 12 | **4** |
| Strongly Correlated | 20 | 1.604 | **1.368** | 5 | **1** |
| | 30 | **0.610** | 0.796 | 7 | **2** |
| | 40 | **0.443** | 1.375 | 11 | **3** |
| | 50 | **0.073** | 2.333 | 9 | **2** |
| | 60 | **0.042** | 0.510 | 11 | **4** |
| | 70 | **0.020** | 0.623 | 16 | **3** |
| | 80 | **0.000** | 1.097 | 13 | **3** |

Table 7: Median RE and solving times for knapsack instances with LP relaxation. For each row, the median RE and average solving time are computed over 18 instances. All times in seconds. The smallest (best) values in each row/metric are in bold.

consist of two components that are on entirely different scales. Specifically, the first-stage objective will be tight as it is being represented exactly while the second-stage objective requires the relaxation of the prediction model which will not be tight due to the big-$M$ constraints. This means that the maximization problem in the AP favors the second stage. This mismatch could lead to inaccurate scenarios and undesirable downstream effects within branch-and-bound.

## H.4 BASELINE SOLUTION QUALITY AT NEUR2RO TERMINATION TIME

In this section, we report the objective quality, i.e., the median relative error, for $k$-adaptability baseline at the termination time of Neur2RO in Table 10. From the table, we can see that the performance is median RE of Neur2RO is marginally better than when $k$-adaptability is given 3 hours, except $n = 20, 40$. Note that these tables are only be reproduced for capital budgeting as

| # items | Median RE | | Times | |
|---|---|---|---|---|
| | standard | LP relaxation | standard | LP relaxation |
| 10 | **1.105** | 2.663 | 59 | **4** |
| 20 | **0.000** | 0.060 | 324 | **142** |
| 30 | 0.109 | **0.071** | 602 | **141** |
| 40 | 0.009 | **0.007** | 739 | **226** |
| 50 | **0.001** | 0.001 | 1,032 | **231** |

Table 8: Median RE and solving times for capital budgeting instances with LP relaxation. For each row, the median RE and average solving time are computed over 50 instances. All times in seconds. The smallest (best) values in each row/metric are in bold.

| # items | Median RE | | Times | |
|---|---|---|---|---|
| | sum | second only | sum | second-only |
| 10 | **1.105** | 2.424 | **20** | 233 |
| 20 | **0.000** | 0.192 | **324** | 1,823 |
| 30 | **0.109** | 0.151 | **602** | 3,823 |
| 40 | **0.009** | 0.010 | **739** | 4,062 |
| 50 | **0.001** | 0.005 | **1,032** | 7,424 |

Table 9: Sum and second-stage only predictions for capital budgeting instances. For each row, the median RE and solving time are computed over 50 instances. Note that in these results, the RE is calculated with respect to the $k$-adaptability and each respective ML-approach. All times in seconds.

we do not have the knapsack results throughout the solving process, given only the final objective values are reported in Arslan & Detienne (2022).

| # items | Median RE at 3 hours | | | | Median RE at `Neur2RO` termination | | | |
|---|---|---|---|---|---|---|---|---|
| | `Neur2RO` | $k=2$ | $k=5$ | $k=10$ | `Neur2RO` | $k=2$ | $k=5$ | $k=10$ |
| 10 | 1.105 | 1.140 | **0.000** | **0.000** | 0.809 | 1.559 | **0.267** | 0.359 |
| 20 | **0.000** | 0.196 | 0.112 | 0.064 | **0.011** | 0.240 | 0.098 | 0.084 |
| 30 | 0.109 | **0.020** | 0.073 | 0.032 | 0.102 | 0.067 | 0.093 | **0.029** |
| 40 | **0.009** | 0.074 | 0.011 | 0.019 | **0.013** | 0.079 | 0.058 | 0.019 |
| 50 | **0.001** | 0.033 | 0.039 | 0.020 | **0.002** | 0.035 | 0.006 | 0.008 |

Table 10: Median RE for capital budgeting at 3 hour time limit and `Neur2RO` termination time. For each row, the median RE and average solving time are computed over 50 instances. All times in seconds. The smallest (best) values in each row/metric are in bold.

# I  MACHINE LEARNING MODEL DETAILS

## I.1  FEATURES

Here we provide the features for each of the problems. In both cases, set-based architectures Zaheer et al. (2017) with parameter sharing utilized, so we report the features for a single dimension of the first-stage decision and scenario accordingly. Table 11 reports all of the features for each instance.

## I.2  MODEL HYPERPARAMETERS

This section reports the hyperparameters for the neural networks for each problem. For both problems, we have the same architecture with slightly different hyperparameters. As the objective of `Neur2RO` is to enable efficient optimization, we train small networks that can achieve a low mean absolute error value to ensure that the main and adversarial problems are tractable. For this reason, no systematic hyperparameter tuning was done. Hyperparameter optimization would likely only further improve the already strong numerical results. For both problems, we train a model for 500

| Problem | First-Stage Features | Scenario Features |
|---|---|---|
| Knapsack | $x_i, f_i, \bar{p}_i, \hat{p}_i, r_i, c_i, t_i, C$ | $\xi_i, f_i, \bar{p}_i, \hat{p}_i, r_i, c_i, t_i, C$ |
| Capital budgeting | $x_i, r_i, c_i$ | $\left(1 + \mathbf{\Phi}_i^\mathsf{T}\boldsymbol{\xi}/2\right)_i, \left(1 + \mathbf{\Psi}_i^\mathsf{T}\boldsymbol{\xi}/2\right)_i, r_i, c_i$ |

Table 11: Features for first-stage decision and scenario embedding networks.

epochs and compute the mean absolute error on a validation set every 10 epochs. We then use the model with the lowest reported mean absolute validation error during training for evaluation.

Table 12 reports the hyperparameters for each model. As our model generalizes across instances, which requires invariance to the order and number of decision variables, both the first-stage and scenario embedding networks are set-based architectures (Zaheer et al., 2017). We refer to Figure 3 for a refresher on the overall architecture which has the following hyperparameters. The hyperparameters "$\hat{\Phi}_\mathbf{x}$ dimensions" and "$\Phi_\mathbf{x}$ dimensions" correspond to the hidden and embedding dimensions of the first-stage embedding network. Specifically, "$\hat{\Phi}_\mathbf{x}$ dimensions" corresponds to the network with shared parameters that embed the representation for each first-stage decision. The last dimension of "$\hat{\Phi}_\mathbf{x}$ dimensions" is that of the aggregated vector. The hyperparameter "$\Phi_\mathbf{x}$ dimensions" corresponds to the network that takes the aggregated first-stage embedding vector as input. The last dimension of "$\Phi_\mathbf{x}$ dimensions" specifies the embedding dimension of the first-stage embedding network. "$\hat{\Phi}_{\boldsymbol{\xi}}$ dimensions" and "$\Phi_{\boldsymbol{\xi}}$ dimensions" are analogous for the scenario embedding network. "$\Phi$ dimensions" correspond to the hidden dimensions of the value network. Finally, "aggregation type" specifies the type of aggregation that combines the first-stage/scenario embeddings.

| Hyperparameter | Knapsack | Capital budgeting |
|---|---|---|
| Feature scaling | min-max | min-max |
| Label scaling | min-max | min-max |
| # epochs | 500 | 500 |
| Batch size | 256 | 256 |
| Learning rate | 0.001 | 0.001 |
| Dropout | 0 | 0 |
| Loss function | MSELoss | MSELoss |
| Optimizer | Adam | Adam |
| $\hat{\Phi}_\mathbf{x}$ dimensions | [32, 16] | [16, 4] |
| $\Phi_\mathbf{x}$ dimensions | [64, 8] | [32, 8] |
| $\hat{\Phi}_{\boldsymbol{\xi}}$ dimensions | [32, 16] | [16, 4] |
| $\Phi_{\boldsymbol{\xi}}$ dimensions | [64, 8] | [32, 8] |
| $\Phi$ dimensions | [8] | [8] |
| Aggregation type | sum | sum |

Table 12: Hyperparameters for neural networks.

## I.3 TRAINING CURVES

Figures 9-10 plot the mean absolute error at every 10 epochs during training for the training and validation data. Generally, the training and validation mean absolute error is very close, and in both problems, a relatively low mean absolute error is achieved.

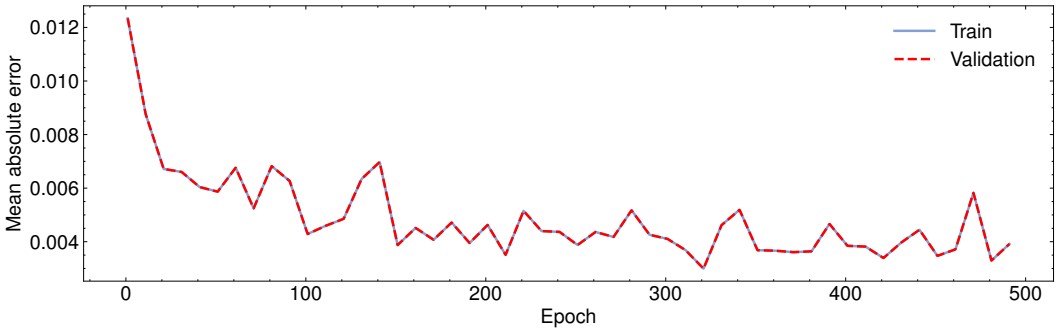

Figure 9: Training curve for knapsack.

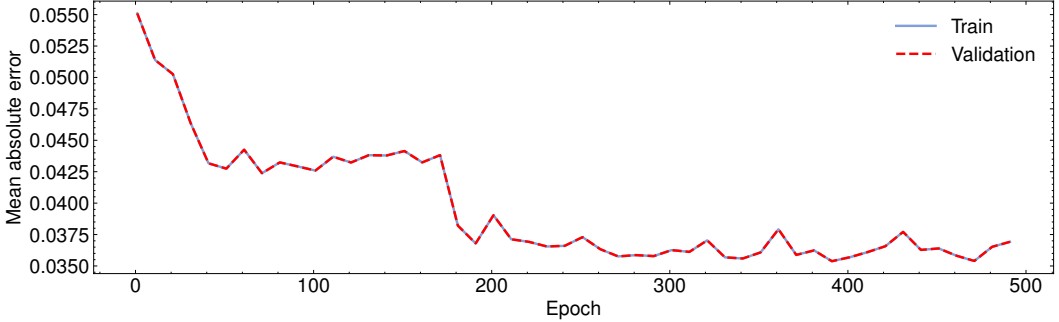

Figure 10: Training curve for capital budgeting.

