# OpenReview forum: "Neur2RO: Neural Two-Stage Robust Optimization"
_ICLR.cc/2024/Conference — ICLR 2024 poster_

### Official Review · Reviewer_fXRT · 2023-10-30

**Soundness:** 3 good
**Presentation:** 4 excellent
**Contribution:** 3 good
**Rating:** 8
**Confidence:** 4

**Summary:**

The authors propose an approach for solving two stage robust optimization that empirically solves previously intractable problems. The setting considers optimization problems where a set of initial decisions are made, and then a worst-case setting is evaluated and in that worst-case setting the decision-maker can make second-stage decisions to react and ideally improve their deployment. Here the first stage solutions are evaluated based on how robust they are to adversarial settings considering that the decision-maker can react in a pre-specified manner. As this problem considers the nesting of 3 optimization problems it’s empirically very difficult to solve. The standard method here is to maintain a main problem, and then add worst-case scenarios modeling them as cutting planes. The issue here is that due to the inherent multi-level optimization problem, these problems are intractable. The proposed approach is to approximate the value of the lower-level problem in each of these settings using a neural network and then embed the neural network in the outer optimization formulation. Here the neural network is trained to take in a first stage decision as well as a setting and output a value estimating the quality of the best possible second stage recourse given the first stage solution and setting. As such, finding the worst-case adversarial setting amounts to optimizing the output of the neural network value estimator, and finding an estimate of the best possible first-stage solution amounts to optimizing over all possible settings, the output of the neural network estimator. They demonstrate performance on a robust two stage knapsack setting and a capital budgeting setting, demonstrating that their approach can give high-quality solutions in tens of seconds which aren’t solvable by previous methods in 3 hours. The approach demonstrates scalability, and the authors also prove convergence guarantees demonstrating that the algorithm is guaranteed to converge in finite settings.

**Strengths:**

The core strengths are the drastically improved performance over previous approaches and well-motivated approach. ¬
The improved performance means that problems that were previously intractable are now heuristically solved in tens of seconds, enabling the use of two stage robust optimization for larger or more complex problem settings.
Additionally, the approach itself is well-motivated by approaches in the optimization literature. By embedding a neural network in the subproblem solver as well as the top-level solver they ensure that the deployed solution is guaranteed to satisfy the constraints and be robust to the enumerated settings. Furthermore, by working with an exact solver, and using the true objective function, the authors guarantee that the resulting objective value of the top-level problem is an objective value corresponding to the first-stage solution for an actual setting.

**Weaknesses:**

The main weakness is that the scope of impact is somewhat limited to two stage robust optimization. It might be helpful to expand somewhat on where this approach might be useful by giving some examples that are two stage robust optimization from the cited survey on adjustable robust optimization. Alternatively, it might be helpful to give some examples on ways that this can be readily extended to cover more flexible problems such as general-purpose multi-level solver.

Additionally, it is not the first use case of embedding neural networks in optimization to solve hard optimization problems as this was previous done in the cited Dumouchelle 2022 paper. However, this is certainly not a direct extension of that work as the neural network is embedded both in the subproblem as well as the main problem in a manner that is cohesive with the optimization formulation to ensure convergence guarantees and which mimics the original solver.

One slight weakness might be that there are limited baselines. However, it should be noted that there are not many general-purpose solvers in these settings. Would it be possible to compare against a naïve extension of Dumouchelle 2022? It might be possible to train a neural network to take in as input only first stage decisions and output the value of all subsequent decisions. It might be computationally expensive to create a training dataset as one would have to evaluate first stage solutions; however, it may be possible. It seems that might be more difficult for a neural network to model, but potentially give a heuristic solution with one solver call.

Another slight weakness is that there are not many settings evaluated outside of two knapsack-like settings. It might be helpful to demonstrate this approach on other two stage robust optimization settings with potentially more complex feasible regions that would potentially be more difficult for the neural network to model.

**Questions:**

It may be helpful to define what the “adversarial problem” is versus the “second stage problem” (and the corresponding second stage value function). It seems that the adversarial problem is the minimization over xi. Is the second stage problem the optimization over y?

Are the samples used for training randomly drawn before any solving happens? Are those samples representative of the settings in which the optimization will be occurring over? For instance, the optimization may be performed over a very different part of the feasible space than what was used for sampling. It would be interesting to see a measure of accuracy for the learned models over time at different stages of the solving process.

What does the solving performance look like over time? The solution qualities are somewhat similar whereas the runtimes are drastically different. Is it the case that the baseline very quickly jumps to high quality solutions and stays there? Or does it continue to improve gradually. You might consider evaluating a form of primal integral measuring the performance of the first stage solutions over time. As a quick solution, you might also consider evaluating the solution quality at around the same time it takes the Neur2RO model to terminate.

It would help to have notions of error on the statistics since some of the relative errors are somewhat close.


Small comments:
P6, experimental setup 2RO Problems
As larger or larger -> as large or larger

---

> ### Author Response · Authors · 2023-11-16
> **Response to Reviewer fXRT**
>
> Thank you for the detailed comments.  We will first comment on a couple of the weaknesses mentioned to provide more insight from our perspective.
> - Firstly, regarding the use of predicting the value function using only the first-stage decision. We have thought about this as well, but in general came to similar limitations as you mentioned above.
>     - As you noted, data collection will be expensive.  In fact, obtaining the regression target (the objective of the inner max/min) requires solving a bilevel optimization problem with integer follower decisions (at least for problems with constraint uncertainty, e.g., capital budgeting), so this may be prohibitively expensive, and will likely be much less scalable to larger instances and other more challenging 2RO problems.
>     - We also agree that it will likely be more difficult to model the max-min function based on the first-stage decision alone.  This difficulty may be even more significant when training across instances with varying sizes, objectives, and feasible regions, as explored in this work, but not [Dumouchelle, 2022].
> - Regarding the benchmarks, both problems indeed involve similar knapsack-like constraints. Our method would, in theory, be capable of handling more than this, as we predict the second-stage objective with respect to its constraints.  We are currently working on an extension of this work that addresses problems with more challenging constraints, namely, a facility location problem. Our first tests indicate that a similar performance boost is possible as shown in the paper. However, due to the limited time we cannot provide these new computational results in this submission.   These additional results will be included in the final version of the paper if accepted, space permitting.
>
> Next, we will address the questions.
> 1. The second-stage problem is purely an optimization over the second-stage variables $y$, where $x$ and the scenarios are fixed (see Equation 3 from the paper). The adversarial problem is the $\max_\xi \min_y$ problem, which is a bilevel integer optimization problem (see Figure 1, box(c) from the paper, or Equation 8 from the supplementary material).
> 2. The samples are indeed drawn randomly over the entire uncertainty set. We have discussed this issue among ourselves, and we have conducted some small experiments along the lines you proposed.
>     - Specifically, we compare the predicted objective (i.e., the max of the NNs) to the actual objective in the main problem.  For strongly correlated knapsack with $n=20$ and $80$, the MAE is $0.021$ and $0.034$, respectively. For capital budgeting with $n=10$ and $50$, the MAE is $0.173$ and $0.184$, respectively.  Generally, these results are at most one order of magnitude worse than the MAE reported during training.
>     - In our opinion, this further justifies the use of argmax as misprediction in the main problem can be mitigated.
>     - It also motivates that if more time is spent on finding worst-case scenarios, either during data collection or using scenarios found during solving to retrain, this would likely only improve the already strong numerical results we have.
> 3. To give more insight into the solving process, see the points below:
>     - Generally, the baselines (k-adaptability) will find decent but worse solutions somewhat early in the solving process and slowly improve over time.
>    - We have added the table comparing the solution quality at ML termination time to the supplementary materials in Table 8.  From the table, we can see that the performance is slightly better at the time of ML termination, but not always, i.e., for $n=20$ median relative at ML time, it is slightly worse.
>     - We would also like to mention that these are the updated results for capital budgeting, for which the improvement over the baseline is more significant for our method.
>     - We agree that comparing primal integrals over time would be a valuable performance measure and will be considered for future extensions of this work.
> 4. See Tables 4-5 and Figures 4-8 in the supplementary materials for statistics and box plots of the relative error.

---

> > ### Comment · Reviewer_fXRT · 2023-11-21
> >
> > The authors adequately address my comments and questions and have provided numerical results to further improve their claim that their approach substantially outperforms the relevant baselines for this problem setting. I have raised my score to accept to reflect this (6->8).

---

### Official Review · Reviewer_viyN · 2023-11-01

**Soundness:** 3 good
**Presentation:** 3 good
**Contribution:** 2 fair
**Rating:** 6
**Confidence:** 4

**Summary:**

In this paper the authors combine machine learning and optimization to develop an algorithm to solve two stage robust optimization problems in a more efficient fashion. They do so by modifying the column and constraint generation algorithm for two stage RO problems.

They train a Neural Network which predicts the optimal value of the second stage problem given the first stage decisions and the uncertainty realizations and incorporate the NN into the mathematical programming problems used to compute the initial decision and uncertainty realizations.

This allows for faster and better computation of the first stage decisions and the worst case realizations.
They also show that their algorithm will terminate after a finite number of steps if the initial feasible region is finite.

They numerically illustrate their results on a Knapsack and a capital budgeting problem.

**Strengths:**

**originality**: The use of neural networks to predict value functions is quite commmon especially in reinforcement learning literature. However, the incorporation of the neural network into a standard mathematical programming problem is new.

**quality**: The algorithm developed is justified theoretically and works well on the problems under consideration.

**clarity**: The paper is well written and presents its arguments well. The experiments are clear and succinct.

**significance**: This approach presents a new approach which can be used to solve challenging two stage constrained robust optimization problem while still maintaining optimality guarantees.

**Weaknesses:**

The applications considered are quite limited and don't really give an idea of how the approach will work in problems with other constraints beyond just the packing constraint.

**Questions:**

1. How much of an impact does the complexity of the Neural Network have on solution time vs relative error.
2. Did you evaluate the approach on any non binary problems. How was the performance on them.
3. Did both the knapsack and the budgeting problems involve only one constraint? Did you try the approach on problems with more constraints?

---

> ### Author Response · Authors · 2023-11-16
> **Response to Reviewer viyN**
>
> Thank you for the review, we will provide answer to the questions in this comment.
>
> 1. We did not compare this systematically.  However, generally, we observed that decreasing the capacity significantly resulted in a training/validation error that was significantly larger, whereas increasing capacity did not significantly improve the error.  In general, larger NNs are more challenging to optimize over in the MILP, as it requires more variables and constraints so that we would expect a significant increase in solution time.  In future work, we will aim to address this in more detail.
>
> 2. The main focus of this work is on two-stage robust optimization (2RO) problems with discrete recourse.  As we have mainly focused on benchmark instances from the 2RO literature, most of these include only binary decisions.  We note that there is nothing limiting our approach to the binary case, so we agree this would be an interesting comparison.
>
> 3.  Both problems do indeed involve one (non-trivial) constraint. Our method would in theory be capable of handling more than this, as we predict the second-stage objective with respect to its constraints.  We are currently working on an extension of this work that addresses problems with more challenging constraints, namely, the facility location problem.

---

### Official Review · Reviewer_3Yd3 · 2023-11-02

**Soundness:** 3 good
**Presentation:** 3 good
**Contribution:** 2 fair
**Rating:** 6
**Confidence:** 4

**Summary:**

The authors study the problem of solving robust mixed integer programming problems using deep learning. As a main contribution, deep neural networks (DNN) are used as a building block to accelerate existing column and constraint generation algorithms. To be specific, the DNN model is first used to learn a mapping from input to the objective value of the stage-2 problem. Then, the trained DNN is embeded into the original robust optimization problem based on its mixed integer linear programming formulation. Simulations on over several cases are conducted to show the effectiveness of the proposed approach.

**Strengths:**

1. The topic of the paper is interesting.
2. The paper proposes a new deep learning approach to solve the robust mixed integer programming problems.
3. The paper is easy to follow.

**Weaknesses:**

1. The contribution of the paper is not clear. See the comments below.

2. The simulations are not sufficient. See the comments below.

**Questions:**

1. The contribution of the paper is not clear. In [R1], deep neural network (DNN) is used to solve the stochastic mixed integer programming problem. The key idea is to use DNN to learn the input to objective value mapping of the stage-2 problem. Then, the DNN model is embeded into the original stochastic programming problem as a mixed integer programming formulation. The idea of the paper is similar to that in [R1]. The authors are suggested to explain the key differecens the these two papers and show the contribution of the paper clearly.

2. In paragraph 4 of Sec. 3.2, the authors mention "Prediction inaccuracy is then compensated for in equation 4a by exactly modeling the second-stage cost. As a result, when solving the MP, the true optimal first-stage decision for the selected scenario will be the minimizer, rather than a potentially suboptimal first-stage decision based on any inaccuracy of the learning model." This is not easy to follow. The authors are suggested to express it more clearly.

3. To accelerat the column and constraint genration algorithm using DNN, one straightforward approach is using DNN to predict the worst-scenario for the stage-2 problem and then add the obtained scenario to the main problem. The authors are suggested to explain why this straightforwrd design will not work. Otherwise, the authors are suggested to use this straightfoward design as a baseline approach to show the superiority of the proposed approach.

4. The authors are suggested to use more real-world problem as the case study. For example, the robust unit commitment problem in power system operation.

5. The authors are suggested to analyze the impact of the DNN prediction errors to the performance (such as convergence and optimality gap) of the proposed approach.



[R1] Patel, R.M., Dumouchelle, J., Khalil, E. and Bodur, M., 2022. Neur2SP: Neural Two-Stage Stochastic Programming. Advances in Neural Information Processing Systems, 35, pp.23992-24005.

---

> ### Author Response · Authors · 2023-11-16
> **Response to Reviewer 3Yd3**
>
> We will address the first two questions in this comment.
> 1.  Firstly, we do acknowledge that the learning of the value function for the second-stage problem is shared.  However, the domains in which the papers focus, i.e., two-stage stochastic programming (2SP) and two-stage robust optimization (2RO), differ substantially and require specific contributions to the case of 2RO.  Below, we summarize the important differences and contributions.
>     - **Architecture**: In [R1], a major contribution is made by architecture, which leads to efficient optimization by learning to aggregate scenarios.  In 2RO, aggregation of scenario information, especially before the first-stage decision is input to the model as in [R1], is not ideal, as the worst-case scenario directly depends on the first-stage decision.  As such, 2RO essentially requires a learning model that predicts based on a single scenario.  While this can be done with a standard feed-forward style architecture, such as the per-scenario architecture in [R1], this will suffer from the same scalability issues as demonstrated in [R1] as optimizing over an entire neural network for each scenario will quickly become intractable as the number of scenarios increases in the main problem (MP).  This highlights the main contribution of the architecture in our paper, which enables efficient optimization in the MP for 2RO via only embedding the Value Network for each scenario.
>     - **Generalization**: One major limitation of [R1] is that the learning models were trained and evaluated on the same instance with varying scenarios, whereas, in this paper, we propose an architecture that can generalize across families of instances representing coefficients of the optimization problems as features, and applying set-based architectures.
>     - **NN approximations within an iterative algorithm**: One major distinction between [R1] and this work is how the NN-based approximations are utilized.  In the 2SP setting, [R1] use NN-based value function approximation to directly approximate the extensive form, which results in a single-level optimization problem.  To the best of our knowledge, our work is the first to address NN-based value function approximation within the context of an iterative algorithm for mathematical optimization.
>     - **The argmax formulation**: As demonstrated in our supplementary results, the obvious choice of using a max over the NN predictions, which would be equivalent to the expectation in the per-scenario approximation of [R1], results in strictly worse performance.  This argmax formulation for the MP provides an alternative (and completely new) way to approach the CCG main problem by leveraging the actual coefficients of the underlying optimization problem, rather than purely relying on the predictive model.  While this exact formulation may be limited to 2RO, we suspect the general idea, utilizing predictions in conjunction with true parameters of the optimization model, may have extensions within various ML-based algorithms for other mathematical optimization settings.
>
> 2. We will clarify this in detail in the final version of the paper and provide a clearer explanation here.
>     - Firstly, the naive usage of the prediction model in the MILP formulation would be to use the output of the predictive model directly in the objective.  This would be similar to the per-scenario approximation in [R1] wherein the average of predictions is used. A maximum over the predictions w.r.t. different scenarios would then be taken in the 2RO case.  This formulation, which we will refer to as max-MP, is detailed in Equation 14 of the supplementary material.  We refer to the formulation using argmax (Equation 4 in the main paper) as the argmax-MP.
>     - Consider the case where both the max-MP and argmax-MP formulations identify the same worst-case scenario, $\xi$.  From this, we know that if $x^\star$ is the minimizer for a particular scenario in the true MP, then the optimal first-stage decision for the argmax-MP will be $x^\star$ as their objectives are identical.  However, for the max-MP formulation, it may be the case that there exists some $\hat{x}$, such that $c(\xi) ^\intercal \hat{x} + NN(\hat{x}, \xi) < c(\xi)^\intercal x^\star + NN(x^\star, \xi)$.  So in this case, misestimation may in fact lead to a suboptimal first-stage decision in the max-MP formulation.

---

> > ### Author Response · Authors · 2023-11-16
> > **Response to Reviewer 3Yd3**
> >
> > 3. While predicting the worst-case scenario may be possible, there are a few important reasons we consider the prediction of the value function.
> >     - The first notable issue with predicting the scenario directly is that the scenario vector is a high-dimensional constrained vector.  Without even considering the hard constraints on the vector, the high dimensionality alone is already a significantly more challenging prediction task.  Furthermore, it is unclear if minimizing MSE on the uncertainty vector would be reasonable, or how a feasible vector can be recovered if the model predicts an infeasible scenario.
> >     - Another challenge with predicting the worst-case scenario directly is that the data collection would be much more costly.  For example, capital budgeting would require solving a bilevel optimization problem to determine the worst-case scenario. In contrast, we only need to solve a single-level deterministic problem, which is generally much easier.
> >     - Lastly, in some 2RO problems, such as the knapsack problem, the uncertainty dimension changes across instances.  This means that if we want to train a single model across variable-sized instances as we do in this work, then we would need a learning model that is able to predict with variable-sized outputs.  While there are some models that can do this, we generally believe that a much more straightforward approach is handling the variable dimensions within the input of the model.
> >
> > 4. We acknowledge that this is a great direction for future work, and we are currently exploring a facility location problem with disruptions and uncertain demands, which is definitely closer to a real-world problem, but we will consider the robust unit commitment problem as a potential case study as well.  For facility location, our preliminary experiments indicate that a similar performance boost is possible. These additional results will be included in the final version of the paper if accepted, space permitting.
> >
> > 5. We agree that this would be an excellent addition to the paper and we will consider some analysis on this in the future.  To briefly comment on this, we note that there is a trade-off between the prediction error, size, and solving time of the optimization model.  For example, networks with much less capacity often have worse prediction errors but require less optimization time, whereas very large networks may have higher prediction errors, but require significantly more optimization time.  We thank you for the comment and agree that we will aim to include this in future extensions of this work as this is an important question for the practical adoption implications of this work.

---

### Author Response · Authors · 2023-11-16
**General Response and Updates**

Firstly, we would like to thank the reviewers for their valuable feedback.  We will address each reviewer individually but will make a general comment here regarding a minor change that led to significantly improved numerical results, in terms of both time and solution quality, for the capital budgeting problem.  Specifically, the median relative errors reported in the submission have been improved for every instance size, in some cases by 2-3 orders of magnitude, with roughly a 5x reduction in solving time for larger instances.

To summarize the difference, for any problem in which the uncertainty is multiplied by the first-stage decisions in the objective, e.g., capital budgeting, we find empirically that predicting the sum of the first- and second-stage objectives yields significantly better solutions.  Methodologically, we suspect this results from unequal tightness in the LP relaxation of the first-stage cost and second-stage prediction in the main and adversarial problems.  We are happy to discuss this further.  Table 3 in the main paper as well as Table 5 and Figure 8 in the supplementary material, have been changed to reflect this.  The discussion and methodology have been slightly modified to reflect these changes.

---

### Meta-Review · Area_Chair_TmUS · 2023-12-06

**Metareview:**

This work proposes a method to solve two-stage robust optimization through deep learning. All reviewers acknowledge its motivation and contributions. The reviewers also suggest conducting experiments on more real-world problems and the scope of impact can be limited to two-stage robust optimization. Nevertheless, the rebuttal resolves the most concerns of the reviewers and one reviewer raises his/her rating from 6 to 8. Overall, I opt for acceptance.

**Justification For Why Not Higher Score:**

Please see the meta-review

**Justification For Why Not Lower Score:**

Please see the meta-review

---

### Decision · Program_Chairs · 2024-01-16

Accept (poster)